# Non-toxic silver telluride colloidal quantum dot mid-infrared photodetector

So Young Eom[1,2], Jin Hyeok Lee[1,2], Haemin Song[1], Suheon Son[1] & Kwang Seob Jeong [1]✉

As demand for sustainable and biocompatible technologies grows, low-toxicity mid-infrared materials, such as silver chalcogenides, have attracted significant interest. Herein, we report mid-wavelength infrared tunable $Ag_2Te$ colloidal quantum dots through a post-growth method starting from short-wavelength infrared $Ag_2Te$ colloidal quantum dots. Using the synthesized $Ag_2Te$ colloidal quantum dots, we successfully fabricate a photodetector covering the full mid-wavelength infrared spectral range (3–5 μm) with an onset wavelength extending to 6.9 μm. At 78 K, the photodetectors exhibit a photoresponse time of 230 ns (rise) and 576 ns (fall). Responsivity varies from $1.9 \times 10^{-3}$ A W$^{-1}$ at 0.02 V to 1.1 A W$^{-1}$ at 0.5 V, depending on the applied bias, and the specific detectivity of the device is $1.2 \times 10^9$ Jones at 0.02 V. The measured noise-equivalent temperature difference of 0.3 K enables us to reliably distinguish temperature variations between 37 °C and 40 °C, directly enabling the diagnosis of fever-level body temperatures.

During the global pandemic, we realized that visualizing thermal energy is critical for public health analysis. Visualizing thermal energy has been of great interest for this pandemic generation in both fundamental science and society. A mid-wavelength infrared (MWIR, 3–5 μm) photodetector, the chip in an infrared camera, is crucial across various fields, including thermal imaging, vibrational spectroscopy for molecular identification and understanding molecular dynamics, environmental pollution analysis, and free-space optical communications[1–8]. Traditional MWIR technologies heavily rely on epitaxial semiconductors such as HgCdTe, which deliver excellent sensitivity but at a large expense in fabrication due to vacuum-based growth, lattice-matching requirements, limiting their scalability and affordability for next-generation systems[9–16].

Colloidal quantum dots (CQDs) have emerged as a promising alternative, offering widely-tunable energy, strong quantum confinement, and low-temperature solution processing—ideal for scalable and cost-effective infrared optoelectronics. In particular, CQDs are compatible with large-area substrates and can integrate with complementary metal-oxide semiconductor (CMOS) backplanes via an efficient coating process[6,17–20].

Mercury telluride (HgTe) CQDs stand out in this context, supporting both interband and intraband photodetection. Interband transitions have enabled monolithically integrated MWIR imagers with CMOS readout integrated circuits (ROICs)[6,18,19]. At the same time, Fermi-level engineering has enabled intraband transitions to reach the long-wavelength infrared (LWIR) and terahertz ranges, with cut-offs of up to ~18 μm[21–24]. These results show the versatility of HgTe CQDs; however, their intrinsic mercury toxicity remains a barrier to large-scale deployment, particularly in consumer and biomedical markets.

Silver telluride ($Ag_2Te$) has emerged as a promising, mercury-free CQD platform. In 2021, we reported $Ag_2Te$ colloidal quantum dots synthesized through a colloidal route exhibiting extended short-wavelength infrared (eSWIR) photoresponse[15]. This was followed by the development of phosphine-free synthetic methods[25], enabling high-performance SWIR detectors and $Ag_2Te$ CQD ink formulations that produce printed photodetectors with an approximate external quantum efficiency (EQE) of 16% at room temperature[26].

Despite this promising progress, $Ag_2Te$ CQD research has so far mainly been constrained to the SWIR regime, with MWIR detection remaining largely unexplored. The MWIR range is especially critical for

[1]Department of Chemistry, Korea University, Seoul, Republic of Korea. [2]These authors contributed equally: So Young Eom, Jin Hyeok Lee. ✉e-mail: kwangsjeong@korea.ac.kr

applications in imaging and sensing, yet no non-toxic CQD system has demonstrated robust and tunable detection in this regime[25,27,28]. Extending the optical window of the $Ag_2Te$ into MWIR would bridge this critical gap, offering both high performance and environmental safety.

Here, we present $Ag_2Te$ CQDs with widely tunable band gaps reaching the MWIR window (3–5 μm). Utilizing a successive nanocrystal growth method, we achieve precise control over nanocrystal size and composition, enabling MWIR bandgap engineering. Comprehensive optical and structural characterization confirms the high crystallinity and stability of the nanocrystals, a particular challenge of the infrared $Ag_2Te$ CQDs. Solution-processed MWIR photodetectors fabricated from these CQDs exhibit high responsivity and detectivity in the MWIR region. Additionally, thanks to the low noise equivalent temperature difference (NETD) of 0.3 K, we successfully resolve temperature between 37 °C and 40 °C, where the fever of the human body is diagnosed. These results establish $Ag_2Te$ as a less-toxic CQD platform enabling photodetection across the entire 3–5 μm MWIR band, providing a route toward scalable and environmentally benign infrared detection technologies.

## Results

### Synthesis of mid-wavelength $Ag_2Te$ colloidal quantum dots

During the synthesis of $Ag_2Te$ CQDs, the nanocrystal size converged to approximately 6.4 nm, which is regarded as an equilibrium size governed by Ostwald ripening. To overcome this intrinsic size limitation, we developed a growth route for $Ag_2Te$ CQDs. According to the Lifshitz–Slyozov–Wagner theory, increasing the reaction temperature accelerates the dissolution and diffusion of smaller particles, allowing the ripening process to attain a larger equilibrium size[29,30]. However, $AgNO_3$ in oleylamine tends to reduce into Ag nanoparticles at high temperatures (>160 °C), which limits this approach. To address this issue, we synthesized SWIR $Ag_2Te$ CQDs at a lower temperature of 130 °C, purified to remove residual species, and re-dispersed in oleylamine. As-synthesized SWIR $Ag_2Te$ CQDs dispersion was then heated to 180 °C, above the initial synthesis temperature, while a metal precursor and reducing agent were introduced to promote growth beyond the conventional equilibrium size, as illustrated in this experiment in Fig. 1a. In this strategy, the reducing agent activates the metal precursor to form more reactive species[31]. Simultaneously, the elevated synthesis temperature (180 °C) induces partial dissolution of smaller SWIR $Ag_2Te$ nanocrystals, supplying additional monomers. These combined effects allow $Ag_2Te$ nanocrystals to exceed the equilibrium size of Ostwald ripening. The $Ag_2Te$ CQDs initially exhibited a SWIR bandgap, but further growth at 180 °C for 2–6 h, with the precursor and reducing agent, produced MWIR $Ag_2Te$ CQDs with a bandgap extending to 4.7 μm.

Figure 1b–g compares the properties of $Ag_2Te$ grown by the post-growth method (MWIR $Ag_2Te$) with SWIR $Ag_2Te$, FT-IR (Fig. 1b, e), TEM (Fig. 1c, f), and PL data (Fig. 1d, g). While the SWIR $Ag_2Te$ shows an absorption peak at 5570 $cm^{-1}$ with a full width at half maximum (FWHM) of 1450 $cm^{-1}$, and the bandgap PL peak at 4900 $cm^{-1}$ with a FWHM of 850 $cm^{-1}$.

The $Ag_2Te$ CQDs obtained through the post-growth process exhibit broadband absorption extending into 5 μm, the MWIR region, confirming MWIR optical response (Fig. 1e). TEM analysis reveals a clear increase in nanocrystal size from a nominal SWIR $Ag_2Te$ of ~6.4 nm to ~10.1 nm following post-growth. Due to the electron-beam sensitivity of SWIR $Ag_2Te$ nanocrystals, TEM images exhibit merged or blob-like features, which limit reliable extraction of quantitative size distributions. Supplementary Figs. 1, 2 present a representative TEM image of $Ag_2Te$ CQDs synthesized with varying post-growth reaction times, showing a spherical morphology and clear size growth. The absorption spectra reveal a consistent red-shift with increasing CQD size. The resulting size-dependent bandgap energy is further analyzed

using a two-band k·p approximation, as shown in Supplementary Fig. 3. The experimental result on the CQDs size aligns well with the k·p approximation model, suggesting the MWIR bandgap energy of the $Ag_2Te$ remains under the quantum confinement.

For the composition analysis, we used X-ray photoelectron spectroscopy (XPS) to analyze the CQDs. The MWIR $Ag_2Te$ composition is maintained, with distinct Ag and Te peaks, indicating a slight Ag excess (Ag: Te = 2.3:1) composition, related to the large surface-to-volume ratio (Supplementary Fig. 4).

The X-ray diffraction (XRD) patterns of MWIR $Ag_2Te$ CQDs are compared with the bulk monoclinic $\beta$-$Ag_2Te$ reference (Supplementary Fig. 5). XRD pattern analysis reveals a clear monoclinic structure aligned with the reference of PDF#81-1985. The bulk $Ag_2Te$ generally exists in a monoclinic structure below 145 °C, and shifts to a face-centered cubic structure at higher temperatures over 145 °C. Interestingly, in our research, despite using a synthesis temperature of 180 °C—above the bulk transition point—the monoclinic structure still appears. This is consistent with the results of Norris and his coworkers[32], showing that monoclinic $Ag_2Te$ nanocrystals remain stable even when produced at 200 °C.

We investigated the infrared PL spectra from the LWIR (1000 $cm^{-1}$) to the SWIR (8000 $cm^{-1}$) at various temperatures using the home-built cryo-FTIR fluorometer. Interestingly, no signal was observed at room temperature. Many factors can suppress radiative recombination, particularly in the MWIR regime, where the fingerprint vibrational modes of molecules also serve as nonradiative recombination paths. Also, phonons can serve as a non-radiative path for carrier relaxation. To suppress possible nonradiative paths and identify the PL feature, we performed cryogenic MWIR PL spectroscopy. The cryogenic PL measurement reveals that the bandgap PL intensity increases significantly by lowering the temperature (Supplementary Fig. 6). The bandgap PL appears at -2800 $cm^{-1}$ (FWHM = 967 $cm^{-1}$) at 78 K (Fig. 1g).

Interestingly, nonradiative recombination via electronic-to-vibrational energy transfer (EVET) is distinctly independent of temperature, in contrast to phonon coupling and trap-assisted recombination processes, which are discussed in detail in the Supplementary Fig. 6. The EVET rate can be described, following Fermi's golden rule, as[33,34]:

$$k_{EVET} = \frac{2\pi k \kappa^2}{\hbar d_0^2 n^4} \cdot \frac{|\mu_{el}|^2 |\mu_{vib}|^2}{\gamma R^4} \tag{1}$$

where, $k_{EVET}$, $k$, $\kappa$, $d_0$, $n$, $\mu_{el}$, $\mu_{vib}$, $\gamma$, and $R$ are the energy transfer rate, number of accepting modes on a given molecule, geometric factor, effective radius, refractive index, dipole moment of QD, dipole moment of vibrational mode, 1/density of states, and distance.

This suggests that not the temperature but the distance between the crystal domain of the nanocrystal and the vibrational center (IR chromophore) of ligands contributes to the energy transfer rate. Therefore, as reported in previous studies, the EVET with characteristic times ranging from a few picoseconds to several nanoseconds exhibits minimal variation in relaxation time with temperature, making it challenging to detect the PL before relaxation when using the commercial infrared photodetector[35,36].

In Fig. 1g, Supplementary Fig. 7, the vibrational functional group, $CH_2$, of the oleylamine ligand at the QD surface (2850 $cm^{-1}$, $CH_2$ symmetric stretching; 2920 $cm^{-1}$, $CH_2$ asymmetric stretching) efficiently quenches the PL regardless of temperature, which can be understood as a characteristic behavior arising from the EVET.

Sometimes, it is not reasonable to assign the optical absorption feature to electronic states of QDs due to light scattering, vibrational modes, geometrical issues in measurement, etc. Therefore, to demonstrate the electronic transitions, we additionally performed the FT infrared photocurrent spectroscopy covering the LWIR-SWIR

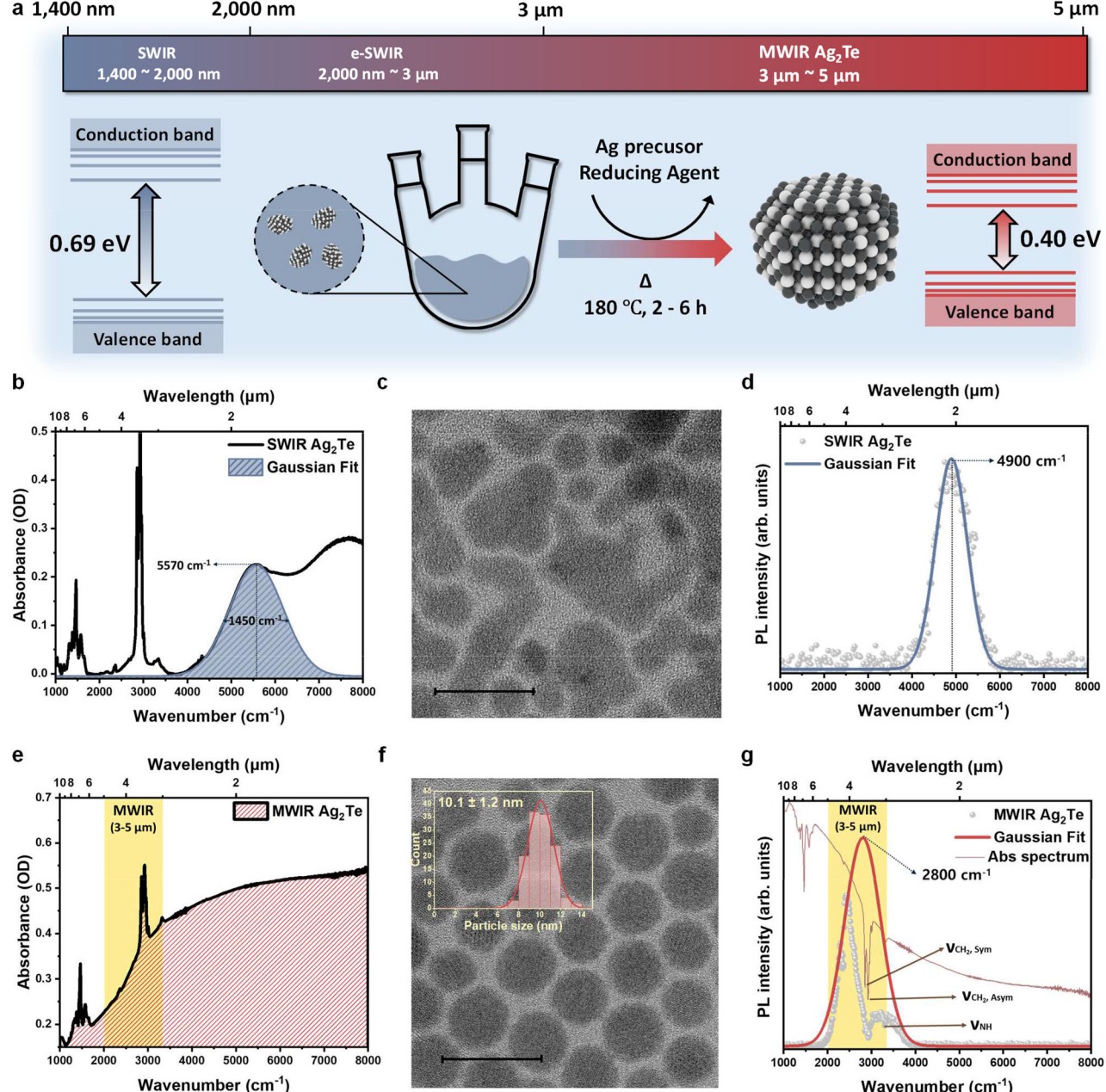

**Fig. 1 | Synthesis schematic and optical characterization of Ag2Te CQDs.**
**a** Schematic illustration of the synthetic process toward MWIR Ag₂Te colloidal quantum dots (CQDs). **b**, **e** Fourier-transform infrared (FT-IR) spectra, **c**, **f** Transmission Electron Microscope (TEM) images (scale bar = 20 nm), and **d**, **g** photoluminescence (PL) spectra of SWIR- and MWIR-Ag₂Te CQDs, respectively.

regime. To measure the photocurrent spectrum, we fabricated the MWIR QD photodetector using as-synthesized Ag₂Te CQDs.

## Mid-wavelength photocurrent spectrum

The photocurrent spectra for the Ag₂Te CQD photodetectors were successfully measured using the home-built FT infrared photocurrent spectroscopy setup. Additionally, we measured the infrared photocurrent spectra from 78 K to 298 K. The blackbody IR source, passed through a Michelson interferometer, was directed at the fabricated CQDs photodetector. The photocurrent signal was processed by fast Fourier transformation, referenced to the He−Ne laser signal, to obtain the photocurrent spectrum as a function of wavelength.

To determine the exact absorption range, the blackbody radiation at 1200 K is irradiated to the photodetector (Fig. 2a). The FT-IR emission spectrum of the blackbody at 1200 K measured by a commercial

MCT detector is in Supplementary Fig. 8. The collected photocurrent spectrum of Ag₂Te photodetector was then used to calculate the relative intensity with the following equation:

$$\text{Relative Intensity} = \frac{\text{Photocurrent of Ag}_2\text{Te CQDs photodetector}}{\text{Photocurrent of commercial MCT photodetector}}$$

$$(2)$$

The relative intensity spectrum of MWIR Ag₂Te shows the infrared-sensitive spectral range from approximately 2300 cm⁻¹ to over 6000 cm⁻¹ for the 3.7 μm Ag₂Te CQDs (Fig. 2b, c). Additionally, the larger MWIR Ag₂Te CQDs have broadened absorption up to 4.7 μm and an absorption onset at 6.9 μm (Fig. 2d). Their bandgap closely matches that of HgTe CQDs, a representative MWIR CQD

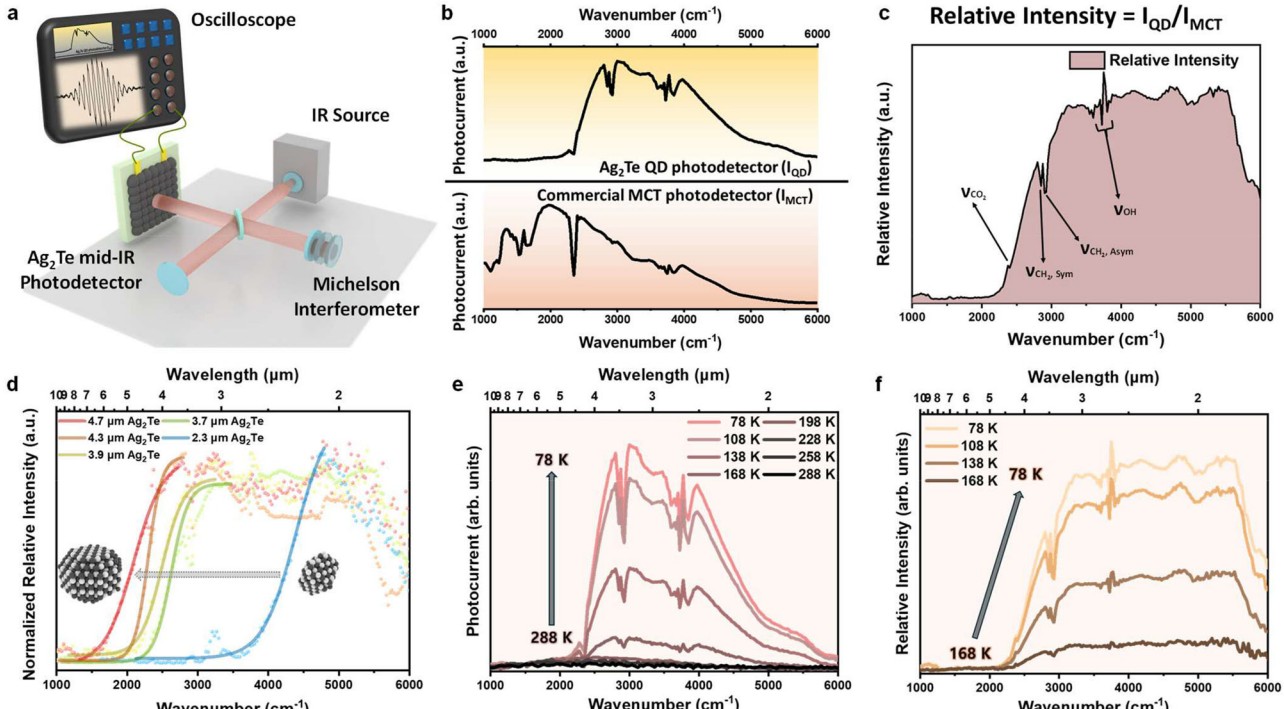

**Fig. 2 | Photocurrent characterization of MWIR Ag2Te CQDs. a** Schematic of the FT-IR photocurrent spectroscopy setup. **b** Photocurrent spectra of the fabricated Ag₂Te CQD photodetector ($I_{QD}$, top) and a commercial MCT photodetector ($I_{MCT}$, bottom). **c** Relative intensity ($I_{QD}/I_{MCT}$) of the photocurrent spectra. **d** Photocurrent spectra of MWIR Ag₂Te CQDs with various bandgap energies tuned by size increase. **e** Temperature-dependent photocurrent spectra of MWIR Ag₂Te CQDs. **f** Relative intensity extracted from the photocurrent spectra at different temperatures from 168 K to 78 K.

(Supplementary Fig. 9). The temperature-dependent photocurrent spectra (Fig. 2e, f) exhibit features corresponding to vibrational modes of CH₂ ligands and atmospheric H₂O (3500–4000 cm⁻¹, O–H stretch) measured under ambient conditions. While ligand exchange with shorter ligands was employed, complete removal of all alkyl species cannot be strictly assumed, and residual vibrational contributions may therefore modulate the MWIR photocurrent response. Additionally, for the MWIR Ag₂Te with the most significant red shift, the bandgap energy widens to include the vibrational energy of atmospheric CO₂, generally less than 800 ppm in a room, confirming its potential for gas sensing applications, particularly for CO₂ detection. This work demonstrates broadband photodetection spanning the entire 3–5 μm mid-wavelength infrared band in silver chalcogenide quantum dots.

By analyzing the relative photocurrent intensity, we specifically observe the infrared light sensitivity within the mid-IR range compared to the MCT detector, along with a consistent decrease in efficiency as temperature rises. This demonstrates that less-toxic materials, such as Ag₂Te, can effectively function as a MWIR sensing material. This capability, which was previously limited to the eSWIR range, is now achieved through an additional growth method.

### Device performance characterization

Figure 3a shows the *J–V* characteristics of the MWIR Ag₂Te CQDs on the Pt Interdigitated Electrode (IDE) device, measured at various operating temperatures ranging from 78 K to 298 K. The corresponding linear fits and calculated method of responsivity for the temperature-dependent analysis are provided in Supplementary Fig. 10. The corresponding responsivities at various operating temperatures are summarized in the inset. At a detector temperature of 78 K and an applied bias of 0.2 V, a responsivity of 0.3 A W⁻¹ was measured. Furthermore, a comparison between the current under true dark conditions and under illumination reveals a difference of ~10³. This clear distinction emphasizes the high sensitivity of the Ag₂Te

device, even in the presence of thermal background radiation at ambient temperatures.

Additionally, the activation energy of the MWIR Ag₂Te material was calculated based on the *J-V* curve data measured at various temperatures. The difference in current values with and without the light source, under varying temperature conditions, was used to calculate the activation energy, as follows.

$$\ln(R_0 A(T)) = \ln(R_0 A(0)) - \frac{E_a}{k_B T} \tag{3}$$

where $R_0 A(T)$, $R_0 A(0)$, $E_a$, $k_B$, T the temperature-dependent resistance–area product from $R_{diff} = (\frac{dV}{dJ})$ at 0.5 V, the resistance–area product at 0 V, activation energy, Boltzmann's constant, and temperature, respectively.

The activation energy, 67.9 meV provides insight into thermally activated carrier generation and transport processes shown in Fig. 3b. A larger activation energy infers the reduced defect-assisted conduction and the suppressed dark current from thermally generated carriers, which is beneficial for device performance.

As we assumed carrier transport mainly through hopping in the CQD films, we analyzed the conductance by varying the temperature. The hopping rate of the MWIR Ag₂Te was determined from temperature-dependent dark-current measurements using Mott's Law[34,37–39]:

$$G \propto \exp\left(-\frac{T_M}{T}\right)^{1/(D+1)} \tag{4}$$

where $G$, $T_M$, and $D$ are the conductance, the Mott characteristic temperature, and the dimensionality (1, 2, or 3), respectively. In this study, we analyzed the low-temperature variation of conductance

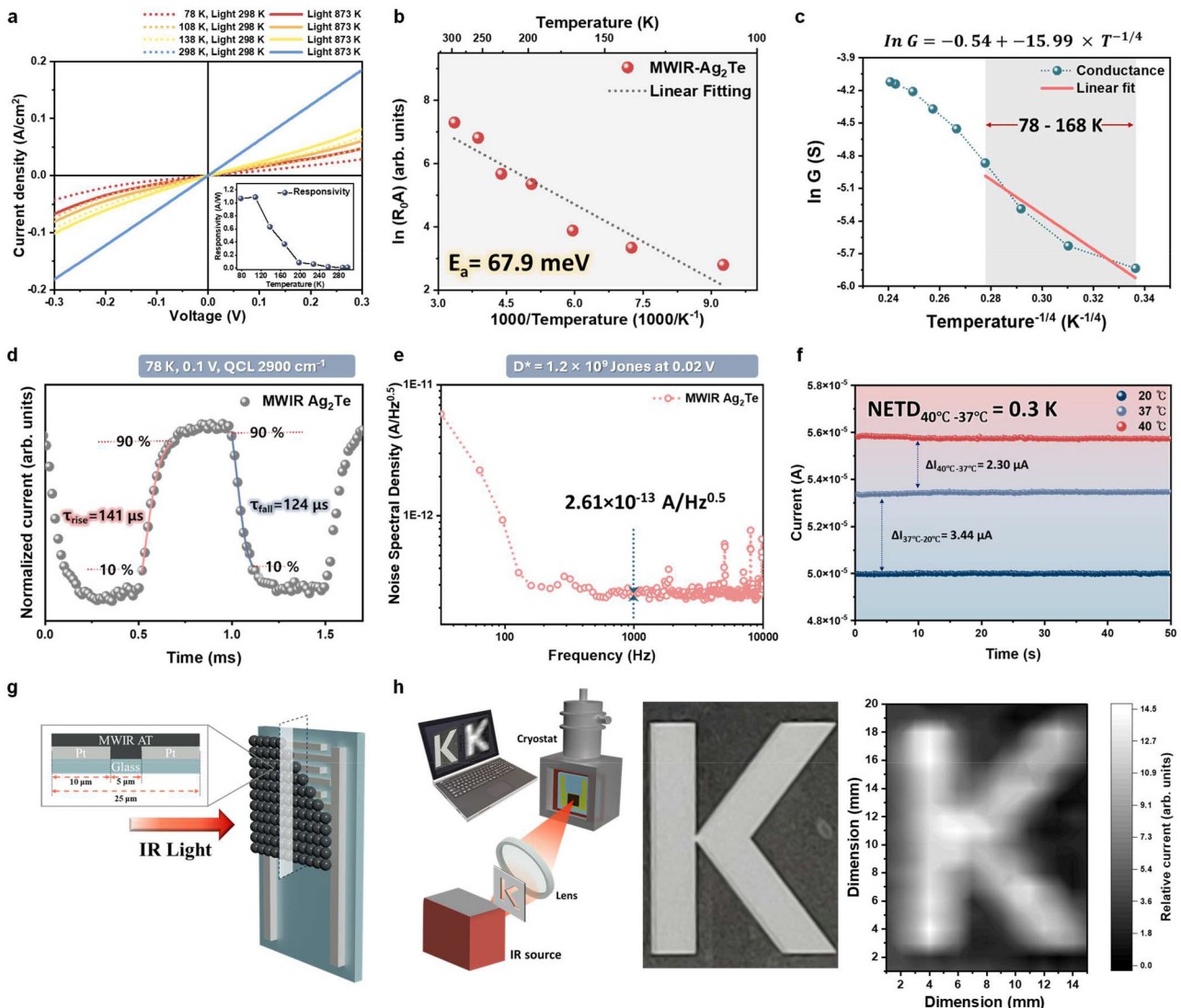

**Fig. 3 | Device performance of MWIR Ag2Te photodetectors (Interdigitated electrode). a** $J$–$V$ characteristics measured at various temperatures. (inset: responsivity versus operating temperature). **b** Activation energy extracted from temperature-dependent measurements. **c** Conductance G as a function of $T^{-1/4}$. **d** Temporal photoresponse at 1 kHz under 2900 cm⁻¹ laser by Quantum Cascade Laser (QCL). **e** Noise current density recorded under dark conditions. **f** Time-dependent current curve at 0.1 V derived from blackbody radiation at 20, 37, 40 °C. **g** Schematic structure of the MWIR Ag₂Te device integrated on the IDE. **h** Infrared thermal imaging setup and corresponding images confirming the photoresponse of the MWIR Ag₂Te device.

using the 3D Mott variable-range hopping (VRH) model (Fig. 3c, Supplementary Fig. 11).

The average hopping frequency at 78 K was then calculated using the Miller–Abrahams hopping formalism[40,41]:

$$f_{hop}(T) = v_0 \exp\left[-\left(\frac{T_M}{T}\right)^{\frac{1}{4}}\right] \quad (5)$$

where the $v_0$ is the hopping attempt frequency (phonon frequency). Assuming $v_0 = 10^{12}\,s^{-1}$ based on the reported paper[40], the Miller–Abrahams expression gives an average hopping rate $4.6 \times 10^9\,s^{-1}$, corresponding to a mean hopping time of $\tau_{hop}(T) = 220\,ps$. (Taking $v_0 = 10^{13}\,s^{-1}$ then the $\tau_{hop}(T) = 22\,ps$).

These findings confirm the rapid modulation, as shown in Fig. 3d, Supplementary Fig. 12. Under illumination by the QCL at 2900 cm⁻¹, the device exhibited clear on–off photoresponses at a modulation frequency of 1 kHz, with a rise time of 141 µs and a fall time of 124 µs. This can be attributed to the efficient generation and the rapid extraction of carriers under illumination. Under cryogenic conditions,

the noise level was measured to be 0.26 pAHz⁻¹/² at 1 kHz shown in Fig. 3e, as detailed in Supplementary Fig. 13. Using this value, the specific detectivity was calculated with the following equation,

$$D^* = \frac{R\sqrt{A}}{i_n} \quad (6)$$

where $R$, $A$, and $i_n$ are the responsivity, device area, and noise current density, respectively.

The device exhibited a specific detectivity ($D^*$, 1 kHz) of $1.2 \times 10^9$ Jones at 78 K under a bias of 0.02 V, with a measured responsivity of 1.9 mA W⁻¹ for a device area of 2.6 mm². This $D^*$ value is comparable to those reported for early HgTe CQDs MWIR photodetectors in 2011[42], indicating that the present device architecture maintains reliable MWIR detection performance. In addition, we summarize and compare the detectivity of previously reported non-toxic MWIR CQD photodetectors, as well as the Ag₂Te CQD photodetector, in Supplementary Table 1. Within this context, the performance of our Ag₂Te photodetector demonstrates a clear advancement among non-toxic

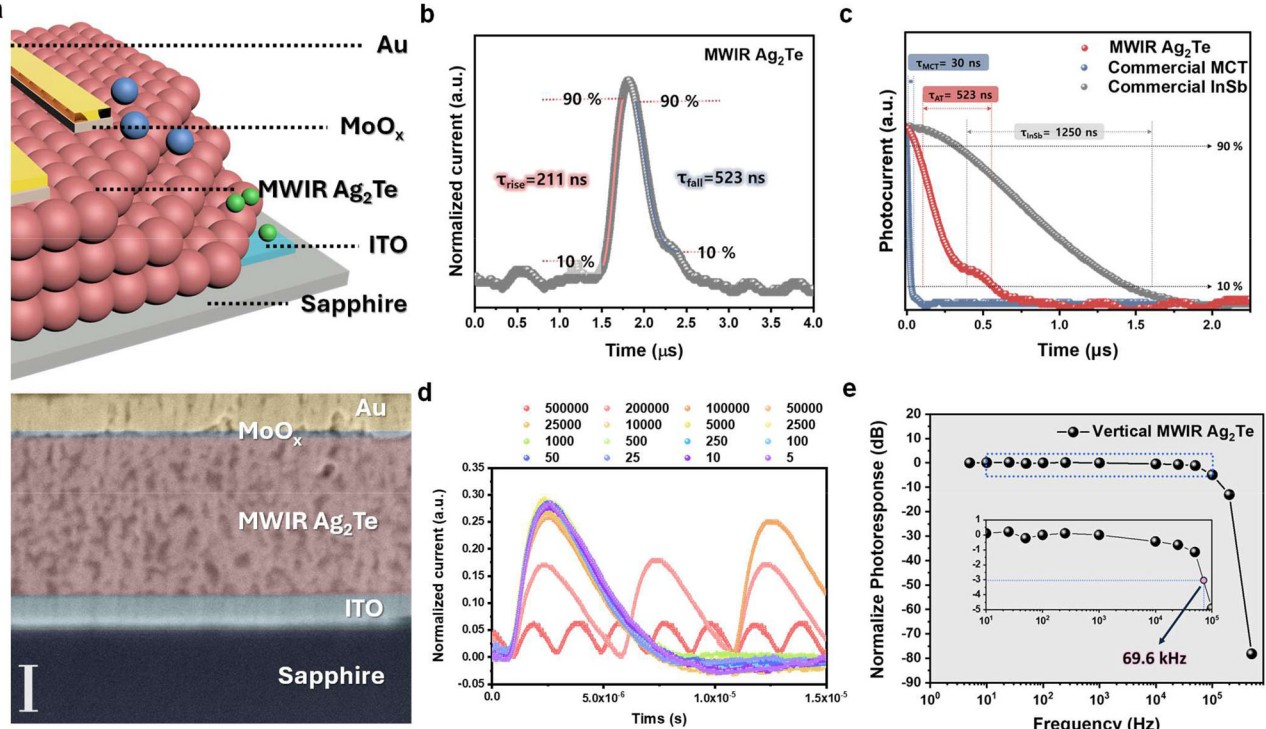

**Fig. 4 | Device performance of MWIR Ag2Te photodetectors (Vertical structure). a** Schematic of the MWIR Ag$_2$Te photoconductor device and a cross-sectional focused ion beam (FIB) image (scale bar = 100 nm) **b** On-off characteristics with a 2900 cm$^{-1}$ mid-infrared QCL. The pulse duration was 625 kHz. **c** The response time of the MWIR Ag$_2$Te CQDs photodetector plotted with those of a commercial MCT bulk crystal photodetector (Blue, Kolmar Tech., KMPV11-1-J1) and

a commercial InSb bulk crystal photodetector (Gray, Infrared Associates, 1176-2C-14-0.5). **d** Time-domain photocurrent responses measured under QCL pulse excitation with repetition frequencies ranging from 5 Hz to 500 kHz. **e** Corresponding normalized photoresponse as a function of modulation frequency, exhibiting a −3 dB cutoff frequency of 69.6 kHz.

MWIR CQD systems reported to date. To show the potential applicability of the device for thermal sensing, it was evaluated under controlled temperatures of the blackbody source. For this purpose, the blackbody temperature was adjusted to represent ambient temperature, normal human body temperature, and a fever-induced body temperature (20, 37, 40 °C), and the corresponding current response was recorded over several tens of seconds (Fig. 3f). A distinct difference in photocurrent values was detected, allowing the calculation of the NETD using the following formula:

$$NETD = \frac{\triangle T}{SNR} = \frac{\triangle T}{\triangle J_s / J_n} \tag{7}$$

where $\triangle T$ represents the temperature difference, $\triangle J_s$ is the measured photocurrent density difference between two blackbody radiation intensities, and $J_n$ is the RMS noise derived from the values at each temperature. From the photocurrent difference observed between 37 °C and 40 °C, the $J_n$ was calculated to be 0.24 μA, and NETD was determined to be 0.3 K, meaning this device can differentiate temperature changes greater than 0.3 K. These results infer potential for further improvement in the millikelvin (mK) temperature resolution through dark current suppression.

Additionally, we took the infrared image using the as-fabricated MWIR Ag$_2$Te single detector (Area = 2.6 mm$^2$, IDE). The image was collected by scanning a thermal image (Fig. 3g, h). The blackbody radiation transmitted through the mask aperture was incident on the fabricated detector, and the resulting photocurrent from each pixel was recorded to reconstruct the infrared image. A Ge window, the long pass filter, was placed in front of the cryostat to block radiation with wavenumber above 5000 cm$^{-1}$, ensuring that the detector responded only to the SWIR and MWIR spectral components of the blackbody

emission. The blackbody temperature was set to 1200 K (Thorlabs SLS303). The results demonstrate the excellent potential of the MWIR Ag$_2$Te CQD-based detector for sustainable IR sensor applications.

Motivated by the estimation of the hopping rate, we also fabricated a vertical-architecture photodetector and assessed whether the intrinsically fast interdot transport can be translated into photoresponse by minimizing the carrier transition length

## Fast response in the mid-wavelength infrared region

A cetyltrimethylammonium bromide (CTAB) passivated mid-IR Ag$_2$Te CQD film was deposited onto a sapphire/ITO substrate, followed by the deposition of MoO$_x$/Au to reduce dark current and utilize the high infrared reflectivity of Au electrode (Fig. 4a). The focused ion beam (FIB) cross-sectional images provide the thickness of the ITO electrode (60 nm), Ag$_2$Te (450 nm) absorbing layer, and uniform MoO$_x$/Au overlays (100 nm).

The vertical device exhibits a measurable photoresponse under blackbody illumination at 873 K, indicating effective infrared photodetection (Supplementary Fig. 14). Under photoexcitation by a QCL at 2900 cm$^{-1}$ (3.4 μm), modulated at 200 kHz, the as-fabricated vertical-type Ag$_2$Te photodetector exhibited a nanosecond-scale photoresponse, with a rise time of 211 ns and a fall time of 523 ns, as shown in Fig. 4b. Such measured response times show competitive performance compared to previous reports (Supplementary Table 2). The measured fall time of 523 ns is the fastest reported for non-toxic MWIR CQDs photodetectors, confirming competitive temporal performance. To compare our results with those of commercial MWIR detectors, we further measured the photoresponse time under the same conditions as shown in Fig. 4c. Surprisingly, the as-fabricated device responds faster than the InSb commercial detector (Infrared Associates, 1176-2C −14-0.5) but slower than the MCT detector (Kolmar Tech., KMPV11-1-

J1). This suggests that the as-fabricated $Ag_2Te$ CQDs photodetector is indeed a promising quantum-dot platform for mid-IR detection. It is important to note that the photoresponse can be influenced by the instrument response function (IRF), which is introduced through an amplifier and results from the amplification of fast transient electrical signals (Supplementary Fig. 15).

To assess the robustness of the device response under repetitive optical excitation, pulse-frequency-dependent photoresponse was measured over a modulation range from 5 Hz to 500 kHz (Fig. 4d). The photocurrent amplitude gradually decreased with increasing repetition frequency, reflecting the finite response dynamics. Figure 4e shows the normalized frequency response yields a -3 dB cutoff frequency of 69.6 kHz. This frequency-domain analysis provides independent validation of the high-speed temporal response, consistent with the trends observed in transient pulse measurements.

These results clearly demonstrate the potential of $Ag_2Te$ CQDs as a competitive sensor material for MWIR detection, with ongoing advances in materials engineering and device design expected to achieve response times in the few-nanosecond range.

## Methods

### Material

Silver acetate (AgAc, Sigma Aldrich, 99%), silver nitrate ($AgNO_3$), tellurium powder (Alfa Aesar, 200 mesh, 99.5%), trioctylphosphine (TOP, Sigma Aldrich, tech. grad. 90%), oleylamine (OLAm, Sigma Aldrich, tech. grad. 70%), diphenylphosphine (DPP, Sigma Aldrich, 98%), Alane N, N-dimethylethylamine (Sigma Aldrich, 0.5 M in toluene), Cetyltrimethylammonium bromide (CTAB, Sigma Aldrich), chloroform (DAEJUNG), and methanol (DUKSAN). All chemicals were used without any purification process.

### Synthesis of SWIR $Ag_2Te$

The 2.4 mmol of AgAc and 24 mL of OLAm were placed in the 100 mL three-neck flask. The solution in the flask was purged with Argon gas for 30 min. The solution became clear brown by increasing the temperature. When the temperature reached 130 °C, 1.2 mL of 1 M TOP-Te was quickly injected. Aliquots were taken and quenched in an ice bath for each desired reaction time. The product solution with chloroform and methanol was centrifuged to remove the residual byproduct.

### Synthesis of MWIR $Ag_2Te$

Pre-formed SWIR $Ag_2Te$ nanocrystals dispersed in OLAm (10 mg/mL) were used as the starting material. The nanocrystal solution was degassed under vacuum (<100 mTorr) for 1 h and subsequently heated to 180 °C under an inert atmosphere. Then, 0.1 mL of 0.1 M $AgNO_3$ solution in OLAm and 0.02 mL of reducing agent (DPP, Alane N, N-dimethylethylamine) were rapidly injected to initiate post-growth. An external reducing agent was added to weaken the bond between Ag+ ions and oleylamine on the surface of nanocrystals to lower the surface energy barrier and introduce additional silver. The reaction was maintained at 180 °C for 2–6 h, during which the nanocrystal size was tuned by varying the post-growth duration. Longer reaction times resulted in larger $Ag_2Te$ nanocrystals and a corresponding red shift of the absorption edge into the MWIR region (Supplementary Figs. 1, 2). After completion, the reaction mixture was cooled to room temperature, and the products were purified by repeated precipitation and redispersion for further characterization and device fabrication.

### Device fabrication (Interdigitated Electrode)

A commercial interdigitated electrode (IDE) with Pt electrodes on insulating substrates was used for lateral photoconductor devices. As-synthesized MWIR $Ag_2Te$ CQDs were dispersed in chlorobenzene at a concentration of -100 mg/mL and deposited onto the IDE substrates by drop casting. After film deposition, a ligand exchange with shorter ligands was performed to improve charge transport within the CQD solid. Specifically, the CQD films were treated with a solution containing the CTAB ligand, followed by rinsing with IPA to remove excess ligand. The resulting CQD films uniformly covered the electrode area, forming an active channel between the interdigitated fingers. Electrical contacts were made directly through the pre-patterned Pt electrodes. The completed devices were loaded into an optical cryostat for temperature-dependent electrical and photocurrent measurements.

### Device fabrication (Vertical structure)

For vertical photodetector devices, sapphire with ITO electrodes (60 nm) on insulating substrates was first cleaned by sequential sonication in acetone and isopropanol, then dried under nitrogen. A film of MWIR $Ag_2Te$ CQDs was deposited onto the ITO electrode by drop casting from a chlorobenzene solution (-100 mg/mL), followed by a post-ligands, identical to the procedure used for IDE devices. After CQD film formation, a $MoO_x$ layer and Au top electrode were sequentially deposited by thermal evaporation to suppress dark current and enhance infrared reflectivity. The final device structure was sapphire/ITO/MWIR $Ag_2Te$ CQD/$MoO_x$/Au.

### Fourier-transform infrared absorption

A Nicolet iS10 FT-IR was used to measure the absorption spectra with a resolution of 0.482 $cm^{-1}$.

### X-ray diffraction

The XRD spectrum was measured by a Rigaku Ultima III X-ray diffractometer with graphite-monochromatized Cu Kα (l = 1.54056 Å). The irradiation power was 40 kV with 30 mA. The spectrum was collected with a 0.01˚ sampling width.

### High-resolution analytical transmission electron microscope

A JEM-F200 (JEOL) model with 200 kV of acceleration voltage and $LaB_6$ of electron source was used for measuring the morphology of SWIR and MWIR $Ag_2Te$ CQDs.

### X-ray photoelectron spectroscopy

The K-alpha Model, which uses the monochromated Al X-ray sources (Al Kα line: 1486.6 eV), was used to collect the XPS spectrum. The X-ray power was 12 kV and 3 mA. The step size of the survey scan and detailed scan was 1 and 0.1 eV, respectively. Based on the value of C 1 s (284.8 eV), the energy calibration was performed.

### Infrared photoluminescence spectroscopy

Researchers built a homemade mid-IR emission spectrometer to measure the photoluminescence of $Ag_2Te$ CQDs. They used a 532 nm PSU-H-LED continuous laser with a 1 kHz chopping rate as the excitation source. The SWIR emission signal was collected by an InSb detector, and an FFT was processed. The samples were prepared by drop-casting onto a sapphire substrate. To conduct temperature-dependent measurements, they used a cryostat (Janis ST-100).

### Infrared Photocurrent Spectra measurement

The photocurrent spectrum was measured using an IR light source (SLS303). The IDA device was placed in a Janis ST-100 optical cryostat, equipped with a $CaF_2$ window, to control the temperature. The device was connected to a current preamplifier (SR570) for output voltage collection, while the sample current was monitored with an oscilloscope (MSO54).

### Photocurrent measurement

The current density-voltage (J–V) spectra were measured under the mid-IR source (Omega BB705). The device was prepared by drop casting on the Interdigitated Electrode, and the ligand was exchanged to a shorter length. For the temperature-dependent J–V measurement, the sample device was loaded into the cryostat (Janis ST-100). The

measured signal was collected by a semiconductor parameter analyzer (Keithley 4200-SCS).

For response time measurements, a Mid-IR Quantum Cascade Laser (Daylight Solutions, 31035-HHG) was modulated using an optical chopper system (Thorlabs, MC2000B) equipped with MC1F60 chopper wheels. The current was amplified by a Low-Noise Current Amplifier (Stanford Research Systems, SR570) and recorded using a semiconductor parameter analyzer (Tektronix MSO54). In the vertical device structure, the quantum cascade laser was operated in pulse-duration mode by 625 kHz.

### Noise equivalent temperature difference measurement

The NETD was calculated from the time-dependent current density spectrum from a semiconductor parameter analyzer (4200A-SCS). A constant bias voltage of 0.1 V was applied to the device, and the blackbody temperatures were set to 20, 37, and 40 °C. For each temperature, the current was recorded over 50 s, yielding approximately 500 data points. The NETD was derived from the current difference between 37 and 40 °C, with the RMS value obtained from the measurements at 37 °C.

## Data availability

All data supporting the findings of this study are available within the paper and its Supplementary Information. Additional experimental data available from the corresponding author upon request.

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

## Acknowledgments

This research was supported by the Basic Science Research Program (RS-2021-NR059609 and RS-2022-NR068167) (K.S.J.). This research was supported by Basic Science Research Program through the National Research Foundation of Korea (NRF) funded by the Ministry of Education (RS-2025-25437404 and RS-2024-00407943) (S.Y.E. and H.S).

## Author contributions

S.Y.E. developed $Ag_2Te$ QD synthesis methods. J.H.L. carried out the temperature-dependent infrared photoluminescence and photocurrent spectroscopy measurement. S.Y.E. and J.H.L. infrared photodetection measurements and analysis. S.S. assisted in synthesizing the QD. H.S. assisted in infrared photodetection. S.Y.E., J.H.L., and K.S.J. wrote the manuscript with contributions from all co-authors, and K.S.J. supervised the project. All authors contributed to the discussion of the experimental results and the manuscript.

## Competing interests

The authors declare no competing interests.
