## [Transparent Peer Review file · Nature Communications]

Non-Toxic Silver Telluride Colloidal Quantum Dot Mid-Infrared Photodetector

Corresponding Author: Professor Kwang Seob Jeong

Version 0:

Reviewer comments:

Reviewer #1

(Remarks to the Author)

In the manuscript titled “Non-Toxic Silver Telluride Colloidal Quantum Dot Mid-Infrared Photodetector”, Eom et al reports a seeded-growth strategy for extending Ag₂Te colloidal quantum dots into the MWIR range and demonstrates photoconductive detection at 78 K. The topic of heavy metal-free infrared colloidal quantum dots is indeed of strong scientific interest. However, after a detailed evaluation, I find that the current work does not reach the level for Nature Communications. The data are not very solid, and the device performance is also not sufficiently strong. The manuscript also suffers from organizational issues. My specific concerns are as follows.

- (1) The claim of the “These results position Ag₂Te as the first mercury-free CQD platform for MWIR optoelectronics” in manuscript is open to further discussion. Several prior works have demonstrated MWIR photodetection using nontoxic or low-toxicity quantum dots, including SnTe (ACS Sens. 2018, 3, 2087–2094) and Ag₂Se quantum dots (J. Phys. Chem. C 2018, 122, 18161–18167; ACS Appl. Mater. Interfaces 2021, 13, 937–943; ACS Appl. Nano Mater. 2019, 2, 1631–1636). Although the working mechanisms may differ and the performances are also far from satisfactory, these studies clearly establish precedent.
- (2) Many FTIR absorption spectra do not show a well-defined excitonic peak, yet the authors perform single-Gaussian fitting to extract band-edge positions. Such an approach is only justifiable when the size distribution is narrow, the excitonic transitions are well resolved, and there is no obvious overlap between different excitonic states. These conditions are not satisfied here, and therefore the extracted bandgaps are not quantitatively reliable.
- (3) The authors claim a size dispersion below 12%, but the provided TEM histograms suggest visibly larger variation. More rigorous statistics (larger sampling, clear histogram binning, and quantitative polydispersity metrics) would be needed to support claims of narrow size distribution.
- (4) In the detectivity calculation, the manuscript lacks several critical parameters, including the active device area and the exact responsivity values used in the calculation. In addition, the measured background noise level must be provided. It is also unclear how the thermal noise was evaluated at 0.02 V.
- (5) The vertical photoconductor speed measurement is not meaningful. A vertical photoconductor geometry naturally reduces transport length and can produce fast response that mainly reflects device structure rather than intrinsic carrier kinetics. Moreover, the manuscript does not provide even basic performance characterization of this vertical photoconductor, yet presents a response speed measurement without specifying fundamental parameters such as the device area.
- (6) Device performance does not meet the field’s state-of-the-art. The reported detectivity (~10⁹ Jones at 78 K) fall significantly below leading HgTe CQDs (10¹⁰–10¹¹ Jones). The sustainability motivation is understandable, but the performance gap remains too large to justify publication without substantially stronger materials advances.
- (7) The synthetic approach appears to rely on classical Ostwald ripening processes, yet the authors claim a distinct seeded-growth mechanism without providing mechanistic evidence or theoretical support.

Reviewer #2

(Remarks to the Author)

The manuscript by S. Y. Eom et al. reports the synthesis, characterization, and mid-wave infrared (MWIR) photodetector performance of Ag₂Te quantum dots (QDs). The topic is both important and timely, given the increasing market demand for RoHS-compliant infrared semiconductors. Considering that viable semiconductor options become increasingly scarce in the longer-wavelength IR regime, this work is unique and will likely attract broad interest from the Nature Communications

readership.

The reported device metrics—including a responsivity of 1.1 A/W, a noise current of 0.26 pA/√Hz, and a resulting specific detectivity of 10^9 Jones—are attractive. However, additional experimental evidence could further substantiate the claims and strengthen the overall impact of the work.

I recommend publication after the authors address a number of minor and a few major revisions that would help strengthen the discussion and reinforce the conclusions presented in the manuscript.

1. In Figure 1c, the TEM image of the SWIR QD seeds show numerous merged or overlapping “blobs,” with several features appearing larger than 20 nm. If these features are not sintered/fused QDs but are instead particles laying on top of each other, a visible TEM thickness contrast would normally be expected—however, such contrast is not apparent. The major issue is that these larger features are not reflected in the size distribution histogram, which suggests a mismatch between the TEM observations and the quantitative analysis. The authors should clarify the nature of these large features and ensure that the size histogram corresponds to the actual particle size distribution observed in the TEM image.

2. In both the main manuscript (k-p approximation) and Supporting Information S3, the authors use a bulk bandgap value of 0.06 eV for monoclinic β -Ag₂Te. This value should be supported with an appropriate literature reference.

3. In manuscript page 7, the terminology EVET should be spelled out first time it is introduced in the manuscript for clarity.

4. In Figure 3a, the legend currently labels all curves as being measured “under light.” However, the dotted curves appear to correspond to the dark measurements. The authors should confirm this and correct the legend accordingly.

Additionally, the manuscript should more clearly describe how the responsivity was calculated. Was responsivity determined from the difference between dark and illuminated current at a specific bias? What is the incident optical power? These should be stated to allow readers to reproduce or benchmark the reported figures.

Finally, Figure 3a contains many overlapping curves across different temperatures, which makes it difficult to evaluate the data. To improve clarity, the authors should include raw dark and light I–V curves at each temperature in the Supporting Information, plotted in a linear scale, so readers can clearly examine the underlying device behavior (i.e. magnitudes of dark current vs. photocurrent).

5. There are few discussions in the manuscript that are confusing to understand:

(1) On page 11 (lines 210–213), the authors state that effective defect reduction may lead to an activation energy larger than the bandgap. It is unclear how this is physically possible. In a semiconductor with negligible defect states, the dominant thermal activation mechanism should be intrinsic carrier generation across the bandgap, which would yield an activation energy equal to the half of the bandgap $E_g/2$ (i.e. how would the activation energy exceed the bandgap)?

Additionally, the manuscript discusses dark-current suppression due to recombination–generation processes occurring “within the depletion region.” It is unclear what depletion region is being referred to, since the device structure is not presented as a diode or heterojunction that would naturally form such a region.

(2) In page 11, hopping frequency equation, the pre-exponent factor ν_0 of 10^{12} /s is used. The manuscript does not explain/discuss how this value was obtained or justified.

(3) On page 12 (lines 228 onward), the authors state that the measured hopping frequency is related to rapid photoresponse modulation. Could the authors clarify this connection? Typically, the carrier transit time across the channel—determined by the source–drain separation and the carrier velocity (or related hopping mobility)—is the more relevant parameter governing photoresponse speed.

Also on page 12 (lines 230–231), the authors state, “This can be attributed to the efficient generation of carriers and the rapid discharge under illumination.” Could the authors clarify what is meant by “discharge” in this context? Does this imply a capacitive effect in the device, or is another mechanism responsible for the observed behavior?

It is recommended that the manuscript be revised to address and clarify the issues outlined above (1–3).

6. In the time-resolved photoresponse shown in Figures 3d and 4b, the modulation frequency appears too fast relative to the device’s response time; specifically, the light is switched off before the device current reaches a stable, steady-state value during the light-on period. While these measurements capture the ‘true’ photoresponse, the rise and fall time estimates presented in Figure 4b may therefore be inaccurate. It is recommended that the authors include additional data at slower modulation frequencies—where the device current reaches full steady state—for both Figure 3d (lateral device) and Figure 4b (vertical device) in the Supporting Information.

7. The current noise data shown in Figure 3e require a more detailed analysis. At present, the manuscript only reports the noise value at 1 kHz. Could the authors clarify the dominant source of noise? For assembled QD films, this is typically 1/f noise. Additionally, is the 1/f knee around 100 Hz in Figure 3e accurate, or could the measured noise level have reached the instrument noise floor (approximately 0.5 pA/√Hz for this type of setup) starting near 100 Hz? Providing a more thorough

discussion or additional measurements would strengthen the presented data.

8. On page 14 (lines 272–273), the authors state that an atomic ligand was used. The specific type of ligand should be clearly specified in the manuscript to ensure reproducibility and clarity.

9. Just out of curiosity regarding the Ag₂Te QDs (not requiring a mandatory revision): Some metal chalcogenide QDs, such as HgSe and Ag₂Se, begin to exhibit intraband characteristics as their size increases. Could the current MWIR Ag₂Te QDs presented in this work potentially display intraband behavior as well?

Reviewer #3

(Remarks to the Author)

In this manuscript, Eom et al. reported a synthetic method for large Ag₂Te QDs with absorption edge extending into MWIR regime, and demonstrated photoconductive detectors exhibiting photoresponse under MWIR illumination. The extension of Ag₂Te QD absorption into the MWIR is potentially interesting. However, in its current form, the manuscript does not provide sufficient mechanistic insights into the growth process, nor does it convincingly advance the understanding of device physics. The work is therefore more of an incremental materials/demonstration paper and of preliminary nature rather than a comprehensive study suitable for Nature Communications.

Below are specific comments:

1. In Fig. 1c, the TEM images of the seed QDs showed clear agglomeration, whereas in Fig. 1f the larger QDs appeared well separated and more uniformly dispersed. The authors should provide a clear explanation for this discrepancy. Is it due to differences in surface chemistry, ligand coverage, sample preparation for TEM, or actual differences in colloidal stability between seeds and grown QDs?
2. The TEM image of the seeds is not shown at an appropriate magnification to reliably determine the average size. A low-magnification image is required to assess the overall morphology. While the seed absorption spectra show distinct transitions, the larger dots do not exhibit clear excitonic features despite appearing size-uniform in TEM. The authors should provide a clear explanation for this discrepancy.
3. The role of the reducing agents is not clearly elucidated. The authors stated that oleylamine can reduce Ag⁺ to Ag nanoparticles, yet they intentionally introduced an additional reducing agent. This raises the concern that metallic Ag nanoparticles could form during synthesis. The authors should explicitly explain whether Ag nanoparticles are indeed formed under the reported conditions and why an extra reducing agent is necessary if oleylamine already has reducing capability, and how its concentration affects QDs' phase purity and sizes.
4. During synthesis, the authors used reducing agents at high temperature, which can readily reduce Ag⁺ to metallic Ag(0). This may significantly increase the metallic Ag content in the QDs. However, the manuscript does not adequately discuss the presence of Ag(0) or its influence on QD growth and the final product. The authors cite only one reference related to PbSe synthesis, which is not directly relevant considering the different reduction/oxidation potential of Ag and Pb. I suggest including the Ag 3d XPS spectra of the seed QDs together with those of the MWIR Ag₂Te QDs. This comparison would be more informative, as the seed Ag₂Te synthesis does not require a reducing agent, whereas the MWIR Ag₂Te QDs do. Additionally, XRD alone cannot confirm or exclude the presence of a certain fraction of metallic Ag(0) within the QDs.
5. The Gaussian peak fitting presented in Fig. 1e is not convincing. The experimental absorption spectra didn't show a well-defined excitonic peak; rather, they exhibit a broad shoulder and likely to be a tail extending into the MWIR. Moreover, the fitted curve does not overlap with the long-wavelength edge particularly well. The Gaussian fitting presented appears somewhat arbitrary and if one plots the absorption spectra together in the same plot they will overlap to a large extent especially towards lower energies. Moreover there appears to be significant discrepancies between TEM images and size histograms in supporting Fig 1 and 2.
6. In the activation energy analysis, it is not clear how the resistance values were extracted from the JV characteristics. The manuscript should specify the exact procedure used to obtain resistance from JV curves (e.g., from the low-bias slope, linear fits, or another method?). Furthermore, since the devices are photoconductors, it would be more appropriate to present JV curves on a linear scale to assess linearity and possible non-ohmic effects. More importantly, the device structure was not clearly described in the main text. A detailed description of the device stack and geometry must be provided.
7. L224, the authors assigned a value for the hopping frequency without any justification. Since this parameter directly influences the hopping rate and response time, its choice cannot be arbitrary. The authors should justify their choices.
8. L239, the authors claim that their device performance is "comparable" to HgTe QD devices. However, the reported specific detectivity appears to be 2 orders of magnitude lower than state-of-the-art HgTe QD detectors (10⁹ vs 10¹¹ Jones, see: Nat. Photon. 18, 1147–1154 (2024); Adv. Mater. 2025, 37, 2416877). This comparison is therefore not justified in its current wording.
9. Several references are missing (e.g., at L45, L112, L224, L281). All statements that rely on prior work must be properly cited.
10. The Experimental Methods section is insufficiently detailed to allow reproducibility. For example, 1) exact precursor amounts and concentrations used for growing the large-size QDs are not mentioned at all! 2) ligands used for device fabrication are mentioned only as 'shorter length'! 3) the device dimensions are not clearly written.
11. How is the size tunability with the reported synthetic method? The authors should include reaction conditions that control QD sizes, and corresponding size/absorption data.
12. The photocurrent spectra show dips around the region where C-H vibrational modes typically appear. Since the authors replaced long chain ligands with shorter ones, these features should not strongly affect the photocurrent. A clear explanation is necessary.
13. The rise and fall time calculations appear optimistic. The photocurrent does not reach saturation at the reported

modulation frequencies, which makes the extracted values unreliable. A dB vs frequency (Hz) plot is necessary to determine the device bandwidth and speed.

14. The authors report noise only at 1 kHz. However, low-frequency noise ($1/f$ noise) is essential for evaluating detector performance. This must be included.

15. The reported NETD of 300 mK is somewhat high for an MWIR photodetector. The authors used a blackbody source to image the letter "K," but a blackbody emits across the entire spectrum from visible to far-IR. The manuscript must specify the blackbody temperature and indicate whether any filters were used to block the SWIR/NIR/visible components.

16. If quantum confinement exists across the MWIR range, the PL emission peak should shift systematically with particle size. Is it possible to show this?

17. Finally, the IDE device is fabricated on a glass substrate. However, glass absorbs strongly in the mid-IR and can generate thermal contributions to the photocurrent. It would be more appropriate to use a substrate such as CaF_2 or sapphire for MWIR measurements.

In summary, the manuscript has the potential to be published in Nat Comm., however, it currently lacks sufficient mechanistic understanding, rigorous spectral analysis, and complete device characterization, as well as essential experimental details to demonstrate a convincing case. Substantial revisions and additional data/analysis would be required before further consideration.

Version 1:

Reviewer comments:

Reviewer #1

(Remarks to the Author)

The authors have revised the manuscript, and the reviewer would like to recommend acceptance of the current version.

Reviewer #2

(Remarks to the Author)

One final minor comment related to original question #4. "In Figure 3a, the legend currently labels all curves as being measured "under light." However, the dotted curves appear to correspond to the dark measurements. The authors should confirm this and correct the legend accordingly."

In response to this comment, the authors added new Figure S10(c) to present the true dark condition I-V.

The optical power of 873K blackbody in the 3-5 MWIR band is about 10^4 times larger than that of the 300K black body. Hence it is somewhat surprising to see that current density of '78K, Light 298K' data jumps to a similar level to '78K, Light 873K data' (i.e. a much larger separation between these two curves would typically be expected). The Figure S10(c) looks more like a LWIR detector rather than the MWIR detector. There might be more complex physics related to this device behavior. Nevertheless, while this is intriguing and may warrant further clarification in future studies, it does not fundamentally alter the main claims or conclusions drawn in the manuscript.

Overall, the authors have satisfactorily addressed the previously raised comments through clarified figures, additional supporting data, expanded discussion, and appropriate references. The manuscript appears ready for publication in its present form.

Reviewer #3

(Remarks to the Author)

The authors have taken my comments into consideration, they have amended several of the previous claims that they could not be supported by the data and now the MS is technically more rigorous. I feel more comfortable now with its publication. There is however a subtle point that requires further attention: the authors mention responsivity of 1.1 A/W in the abstract and at several points in the manuscript, but when they refer to D^* conditions the responsivity is only 1.9 mA/W, i.e. 3 orders of magnitude lower...is this because of the lower applied bias? Can the authors show a bias dependent R, D^* ? Also is the spectrum of EQE applied bias invariant?

For sure the abstract needs to be amended because as written it implies that R of 1.1 A/W and D^* of $e9$ Jones are simultaneously achieved but it seems that is not the case, so it is somewhat misleading.

Version 2:

Reviewer comments:

Reviewer #3

(Remarks to the Author)

the authors addressed my concerns. the paper is now publishable.

Reviewers' comments:

Reviewer #1 (Remarks to the Author):

In the manuscript titled “Non-Toxic Silver Telluride Colloidal Quantum Dot Mid-Infrared Photodetector”, Eom et al reports a seeded-growth strategy for extending Ag₂Te colloidal quantum dots into the MWIR range and demonstrates photoconductive detection at 78 K. The topic of heavy metal-free infrared colloidal quantum dots is indeed of strong scientific interest. However, after a detailed evaluation, I find that the current work does not reach the level for Nature Communications. The data are not very solid, and the device performance is also not sufficiently strong. The manuscript also suffers from organizational issues. My specific concerns are as follows.

(1) The claim of the “These results position Ag₂Te as the first mercury-free CQD platform for MWIR optoelectronics” in manuscript is open to further discussion. Several prior works have demonstrated MWIR photodetection using nontoxic or low-toxicity quantum dots, including SnTe (ACS Sens. 2018, 3, 2087–2094) and Ag₂Se quantum dots (J. Phys. Chem. C 2018, 122, 18161–18167; ACS Appl. Mater. Interfaces 2021, 13, 937–943; ACS Appl. Nano Mater. 2019, 2, 1631–1636). Although the working mechanisms may differ and the performances are also far from satisfactory, these studies clearly establish precedent.

Answer:

We thank Reviewer 1 for the comment. In the previous manuscript, we intended to emphasize the role of Ag₂Te as a mercury-free colloidal quantum dot platform that enables broadband bandgap MWIR (3-5 μm) optoelectronic operation with device-level performance metrics relevant to practical MWIR systems. We realized that our previous statement might overemphasize our results and overlook prior reports. To clarify our contribution to the field, we have rephrased the statement as follows:

Initial version:

(line 60) These results position Ag₂Te as the first mercury-free CQD platform for MWIR optoelectronics, offering a new path toward scalable, environmentally benign infrared detection technologies.

(line 176) This work demonstrates, for the first time, nontoxic QD-based photodetection spanning the entire 3-5 μm MWIR band.

Revised version:

(line 61) These results establish Ag₂Te as a mercury-free CQD platform enabling photodetection across the entire 3–5 μm MWIR band, offering a new path toward scalable and environmentally benign infrared detection technologies.

(line 183) This work demonstrates, for the first time, broadband photodetection spanning the entire 3-5 μm MWIR band using silver chalcogenide QDs.

(2) Many FTIR absorption spectra do not show a well-defined excitonic peak, yet the authors perform single-Gaussian fitting to extract band-edge positions. Such an approach is only justifiable when the size distribution is narrow, the excitonic transitions are well resolved, and there is no obvious overlap between different excitonic states. These conditions are not satisfied here, and therefore the extracted bandgaps are not quantitatively reliable.

Answer:

We thank Reviewer 1 for this insightful comment regarding the reliability of Gaussian fitting of FTIR absorption spectra that lack well-defined excitonic features. We agree that single-Gaussian fitting is only quantitatively meaningful when excitonic transitions are clearly resolved, the size distribution is sufficiently narrow, and spectral overlap between different excitonic states is negligible. As the reviewer pointed out, these conditions are not strictly satisfied in our spectrum.

In response to this comment, we have removed the single-Gaussian fitting analysis and the corresponding extracted band-edge values from the revised manuscript. The FTIR spectrum is now presented descriptively without a fitting, and the discussion has been revised accordingly to avoid overinterpretation. The key findings include the broadband MWIR photoresponse spanning the entire 3–5 μm range, as well as by photodetection measurements that directly demonstrate MWIR optoelectronic functionality. We believe that this revision improves the rigor and clarity of the manuscript and aligns the analysis more closely with the experimental limitations highlighted by the reviewer.

Initial version:

(link 89) The Ag_2Te , produced through the seeded growth process, exhibits absorption at 3220 cm^{-1} (FWHM 1650 cm^{-1}), indicating mid-IR absorption (Fig. 1e).

Revised version:

(link 90) The Ag_2Te CQDs obtained through the post-growth process exhibit broadband absorption extending into $5\text{ }\mu\text{m}$, the MWIR region, confirming MWIR optical response (Fig. 1e).

(3) The authors claim a size dispersion below 12%, but the provided TEM histograms suggest visibly larger variation. More rigorous statistics (larger sampling, clear histogram binning, and quantitative polydispersity metrics) would be needed to support claims of narrow size distribution.

Answer:

We thank the reviewer 1 for the careful assessment of the TEM size statistics. We agree that claims of narrow size dispersion require rigorous statistics (large sampling, well-defined binning, and quantitative polydispersity metrics). In the initially submitted version, the size histogram for the seed samples may indeed appear broader than the stated value.

We clarify that reliable TEM-based size statistics for seed-stage Ag₂Te QDs are experimentally challenging because the particles are susceptible to electron-beam irradiation. During TEM imaging, we frequently observed beam-induced morphological changes (rounding/coalescence/partial melting-like behavior) that can distort the apparent particle-size distribution and lead to overestimated dispersion. Electron-beam-induced heating and irradiation-driven transformations/damage in beam-sensitive nanocrystals and chalcogenide nanoparticles are well documented in the literature. (*Nano Lett.* **2021**, *21*, 8073-8079)

Initial version:

(line 91) TEM analysis demonstrates the increase of nanocrystal size from 6.4 nm to 10.1 nm, which is about 57.8% larger than the seed Ag₂Te.

c

Revised version:

(line 92) TEM analysis reveals a clear increase in nanocrystal size from a nominal SWIR Ag₂Te size of ~6.4 nm to ~10.1 nm following post-growth (~57.8% increase). Due to the pronounced electron-beam sensitivity of SWIR Ag₂Te nanocrystals, TEM images may exhibit merged or blob-like features, which limit reliable extraction of quantitative size distributions.

c

(4) In the detectivity calculation, the manuscript lacks several critical parameters, including the active device area and the exact responsivity values used in the calculation. In addition, the measured background noise level must be provided. It is also unclear how the thermal noise was evaluated at 0.02 V.

Answer:

We appreciate reviewer 1's comments. In this work, the responsivity used to calculate the detectivity was determined from photocurrent modulation at 1 kHz. The detectivity was evaluated using the noise current measured at the same frequency, ensuring a self-consistent definition of D^* under the device's actual operating conditions.

As noted by the reviewer's comment, the noise spectrum reaches a near-constant noise floor around ~100-200 Hz, and using the noise value in this frequency range would also be acceptable. However, it is common practice in the photodetector literature to evaluate detectivity at higher modulation frequencies, such as 1 kHz or 10 kHz, where the noise spectrum is flat and less affected by low-frequency flicker noise (e.g., Nature Communications 10, 2125 (2019); Advanced Science 7, 2000068 (2020)); Advanced Science 11, 2407453 (2024)) In line with this convention, and because our responsivity was measured at a 1 kHz chopping frequency, we used the 1 kHz noise value to calculate D^* (1 kHz). This clarification has been added to the manuscript.

In addition, as suggested by the reviewer, we consider the possibility that the measured noise could approach the instrumental noise at low frequencies. To address this point, we have included additional data in the Supporting Information comparing the device noise with the background noise of the measurement system. The instrument noise floor was measured by shorting the input of the current preamplifier to ground and was found to be significantly lower than the noise measured with the device. The results confirm that the reported noise values are dominated by the device rather than the instrumentation.

Based on the dark current and the dynamic resistance measured at a bias of 0.02 V, the theoretical noise contributions were estimated.

The shot-noise-limited current noise: $0.23 \text{ pA/Hz}^{0.5}$, $i_{shot} = \sqrt{2qI\Delta f}$

The Johnson–Nyquist thermal noise: $0.19 \text{ pA/Hz}^{0.5}$, $i_{thermal} = \sqrt{4k_B R \Delta f}$

The experimentally measured noise density of $0.26 \text{ pA/Hz}^{0.5}$ at 1000 Hz shows close agreement with

these theoretical limits. This correspondence indicates that at a modulation frequency of 1000 Hz, low-frequency 1/f noise is effectively suppressed and the device operates near its intrinsic noise floor.

Accordingly, we used the measured noise density of $0.26 \text{ pA/Hz}^{0.5}$ for the detectivity calculation, as it provides a more realistic and physically justified assessment of the device performance under operational conditions. The manuscript has been revised to clarify this analysis. Based on these considerations, we have revised the manuscript accordingly and added the corresponding background-noise data to the Supporting Information.

Initial version:

(line 233) Under cryogenic conditions, the noise level was measured to be $0.26 \text{ pA}/\sqrt{\text{Hz}}$ at 1000 Hz. Using this value, the specific detectivity was calculated with the following equation,

$$D^* = \frac{R\sqrt{A}}{i_n}$$

where R , A , and i_n are the responsivity, device area, and noise current density, respectively.

The resulting specific detectivity (D^*) of 1.2×10^9 Jones at 78 K demonstrates competitive performance compared to HgTe photoconductor devices reported in previous reports^{45–48}

Revised version:

(line 244) Under cryogenic conditions, the noise level was measured to be $0.26 \text{ pA}/\sqrt{\text{Hz}}$ at 1 kHz, as detailed in Supplementary Fig. 13. Using this value, the specific detectivity was calculated with the following equation,

$$D^* = \frac{R\sqrt{A}}{i_n}$$

where R , A , and i_n are the responsivity, device area, and noise current density, respectively.

The device exhibited a specific detectivity (D^* , 1 kHz) of 1.2×10^9 Jones at 78 K under a bias of 0.02 V, with a measured responsivity of 1.9 mA/W for a device area of 2.6 mm^2 . This D^* value is comparable to those reported for early HgTe CQDs MWIR photodetectors in 2011 (Nature Photonics 5, 489–493), indicating that the present device architecture maintains reliable MWIR detection performance despite employing a non-toxic MWIR CQD photodetectors, as well as Ag_2Te CQD-based devices, in Supplementary Table 1.

Revised version (addition):

Figure S13. The noise spectral density of the Ag_2Te photodetector was measured under dark conditions. At low frequencies (below ~ 100 Hz), the noise exhibits a characteristic $1/f$ dependence. As the frequency increases, the noise spectrum transitions to a frequency-independent region above ~ 100 Hz, corresponding to a white-noise-dominated regime.

Based on the dark current and the dynamic resistance measured at a bias of 0.02 V, the theoretical noise contributions were estimated.

The shot-noise-limited current noise: $0.23 \text{ pA/Hz}^{0.5}$, $i_{shot} = \sqrt{2qI\Delta f}$

The Johnson–Nyquist thermal noise: $0.19 \text{ pA/Hz}^{0.5}$, $i_{thermal} = \sqrt{4k_B R \Delta f}$

The experimentally measured noise density of $0.26 \text{ pA/Hz}^{0.5}$ at 1 kHz shows close agreement with these theoretical limits. This correspondence indicates that at a modulation frequency of 1 kHz, low-frequency $1/f$ noise is effectively suppressed and the device operates near its intrinsic noise floor.

For comparison, the instrument noise floor of the measurement system was independently measured by shorting the input of the current preamplifier to ground. The resulting background noise level is significantly lower than the noise measured with the device across the entire frequency range of interest, including at 1 kHz. This confirms that the noise used for detectivity evaluation is dominated by the intrinsic device noise rather than by the measurement electronics.

(5) The vertical photoconductor speed measurement is not meaningful. A vertical photoconductor geometry naturally reduces transport length and can produce fast response that mainly reflects device structure rather than intrinsic carrier kinetics. Moreover, the manuscript does not provide even basic performance characterization of this vertical photoconductor, yet presents a response speed measurement without specifying fundamental parameters such as the device area.

Answer:

We thank Reviewer 1 for the comment. To address concerns about response time and device architecture, we added the following clarification and additional data into the revised manuscript.

Some previous studies of SWIR-sensitive Ag_2Te photodetectors demonstrated competitive performance with vertical device architecture. Specifically, we reported a response time of 2.7 μs for ink-processed devices (ACS Materials Letters 2024, 6, 4988–4996). To ensure consistency with these established results and to facilitate a direct performance comparison, we therefore also conducted the experiment with a similar vertical structure in the present study.

Achieving fast response times remains a critical challenge in MWIR photodetector research. While commercial silicon photodiodes and SWIR detectors (e.g., InGaAs) routinely exhibit response times in the picosecond-to-nanosecond regime, conventional MWIR photodetectors are typically constrained to the nanosecond-to-microsecond range. This limitation primarily arises from intrinsic carrier dynamics and material constraints inherent to conventional MWIR systems.

In this context, our results suggest that MWIR-responsive Ag_2Te is a promising alternative material system that could overcome these speed limitations, offering a viable pathway toward high-speed MWIR photodetection.

To further support this conclusion, we have included the J–V characteristics, responsivity, and additional data about photoresponse time in the revised manuscript. In addition, response-time measurements performed under various modulation conditions have been added to provide a more comprehensive evaluation of the device’s temporal behavior.

Initial version:

(line 228) These findings confirm the ability for rapid modulation, as shown in Fig. 3d. Clear on–off responses were observed at modulation frequencies from 100 Hz to 1 kHz under a 600 °C blackbody radiation. This can be attributed to the efficient generation and rapid extraction of carriers under illumination.

(line 279) Under photoexcitation by a quantum cascade laser at 2800 cm^{-1} ($3.6\ \mu\text{m}$), modulated at 625 kHz, the as-fabricated vertical-type Ag_2Te photodetector exhibited a nanosecond-scale photoresponse, with a rise time of 230 ns and a fall time of 576 ns. Such measured response times show competitive performance compared to previous reports.

Revised version:

Fig. 3| Device performance of MWIR Ag_2Te photodetectors. (Interdigitated Electrode, IDE) a, J - V characteristics measured at various temperatures. (inset: responsivity versus operating temperature). b, Activation energy extracted from temperature-dependent measurements. c, Conductance G as a function of $T^{-1/4}$. d, Temporal photoresponse at 1 kHz under 2900 cm^{-1} laser by Quantum Cascade Laser (QCL). e, The noise current density recorded under dark conditions. f, Time-dependent current curve at 0.1 V derived from blackbody radiation at 20, 37, 40 °C. g, Schematic structure of the MWIR Ag_2Te device integrated on the IDE. h, Infrared thermal imaging setup and corresponding images confirm the photoresponse of the MWIR Ag_2Te device.

(line 240) These findings confirm the ability for rapid modulation, as shown in Fig. 3d and Supplementary Fig. 12. Under illumination from the QCL at 2900 cm^{-1} , the device exhibited clear on-off photoresponses at a modulation frequency of 1 kHz, with a rise time of $141\text{ }\mu\text{s}$ and a fall time of $124\text{ }\mu\text{s}$. This can be attributed to the efficient generation of carriers and the rapid extraction of carriers under illumination.

Fig. 4| Device performance of MWIR Ag₂Te photodetectors. (Vertical structure) a, Schematic of MWIR Ag₂Te photoconductor device and a cross-sectional focused ion beam (FIB) image **b**, On-off characteristics with a 2900 cm⁻¹ mid-infrared QCL. The pulse duration was 625 kHz. **c**, The response time of the MWIR Ag₂Te QD photodetector plotted with those of a commercial MCT bulk crystal photodetector (Blue, Kolmar Tech., KMPV11-1-J1) and a commercial InSb bulk crystal photodetector (Grey, Infrared Associates, 1176-2C-14-0.5). **d**, Time-domain photocurrent responses measured under QCL pulse excitation with repetition frequencies ranging from 5 Hz to 500 kHz. **e**, Corresponding normalized photoresponse as a function of modulation frequency, exhibiting a -3 dB cutoff frequency of 69.6 kHz.

(line 301) The vertical device exhibits a measurable photoresponse under blackbody illumination at 873 K, indicating effective infrared photodetection (Supplementary Fig. 14). Under photoexcitation by a QCL at 2900 cm⁻¹ (3.4 μm), modulated at 200 kHz, the as-fabricated vertical-type Ag₂Te photodetector exhibited a nanosecond-scale photoresponse, with a rise time of 211 ns and a fall time of 523 ns. Such measured response times show competitive performance compared to previous reports.

(line 317) To assess the robustness of the device response under repetitive optical excitation, pulse-frequency-dependent photoresponse measurements were performed over a modulation range from 5 Hz to 500 kHz (Fig. 4d). The photocurrent amplitude gradually decreased with increasing repetition frequency, reflecting the device's finite response dynamics. Fig. 3e shows the normalized frequency response yields a -3 dB cutoff frequency of 69.6 kHz. This frequency-domain analysis provides independent validation of the high-speed temporal response, consistent with the trends observed in transient pulse measurements.

Revised version (addition):

Figure S12. Temporal and frequency-dependent photoresponse of the device. (a) Time-domain photoresponse measured under optical chopper modulation, showing distinct on–off switching behavior. (b) Pulse-frequency-dependent photoresponse measured by directly modulating the QCL pulse repetition frequency, illustrating the evolution of the response amplitude as a function of pulse frequency.

From the normalized frequency response, a -3 dB cutoff frequency of 3.2 kHz was extracted. This value is in good agreement with the cutoff frequency estimated from the measured fall time using the following relation:

$$\text{Fall-time and bandwidth relationship: } f_{3 \text{ dB}} \approx \frac{0.35}{\tau_{fall}} = 2.8 \text{ kHz } (\tau_{fall}: 124 \mu\text{s})$$

The bandwidth derived from the pulse measurements is in good agreement with the data shown in Fig. 3d.

Figure S14. (a) Current density–voltage (J-V) characteristics of the device measured in the dark and under illumination. The net photocurrent is shown, calculated as the difference between the two curves. (b) On–off photoresponse was measured using an optical chopper, while the intrinsic response time was evaluated using a pulsed laser because the chopper’s modulation frequency was limited to 1 kHz. (c) Responsivity was calculated from current and power densities. The device’s active area is 1.05 mm². At 1 V (positive bias), the responsivity is approximately 40 mA/W. The lower responsivity relative to the IDE structure is attributed to the ITO electrode’s low infrared transmittance.

(6) Device performance does not meet the field's state-of-the-art. The reported detectivity ($\sim 10^9$ Jones at 78 K) fall significantly below leading HgTe CQDs (10^{10} – 10^{11} Jones). The sustainability motivation is understandable, but the performance gap remains too large to justify publication without substantially stronger materials advances.

Answer:

We thank Reviewer 1 for this comment. Rather than directly comparing our results with the current state-of-the-art HgTe-based MWIR photodetectors, we have revised the manuscript to better contextualize the performance.

Specifically, we clarify that the detectivity achieved in this work is comparable to that reported for the first HgTe colloidal quantum dot MWIR photodetectors demonstrated in the early stage of the field (Nature Photonics 5, 489–493 (2011)). At the same time, we emphasize that our Ag₂Te photodetector significantly outperforms previously reported MWIR photodetectors based on less-toxic, heavy-metal-free quantum dots.

To further clarify the impact of our results, we compare our device with previous reports focusing on less-toxic materials and the spectral limits of Ag₂Te:

- InSb CQDs: Recently reported a D^* of 3.1×10^7 Jones at 3.5 μm (*Nano Lett.* 2025, 25, 13549).
- Ag₂Se CQDs (Intraband Transition): Showed a D^* of 7.8×10^6 Jones without cooling in the MWIR region (*ACS Appl. Mater. Interfaces* 2021, 13, 49043).
- SnTe CQDs: Devices are shown to exhibit mA/W responsivity under MWIR flux; however, intrinsic limitations in D^* due to inverse photoresponse behavior (*ACS Sens.* 2018, 3, 2087).
- Ag₂Te CQDs Spectral Limit: While existing Ag₂Te has shown high D^* in shorter SWIR wavelengths (near the 1300 nm: 3×10^{12} Jones (*Nature Photonics* volume 18, pages 236–242 (2024)), 1700 nm: 9.0×10^{10} Jones (*ACS Materials Lett.* 2024, 6, 4988–4996)). To the best of our knowledge, the longest-wavelength detectivity previously reported for Ag₂Te was measured with a 2004 nm laser, yielding only 1.0×10^7 Jones (*Adv. Sci.* 2024, 11, 2407453).

In contrast, the Ag₂Te photodetector presented in this work successfully extends high-performance operation into the MWIR region, achieving the D^* of 1.2×10^9 Jones. While HgTe-based devices remain a benchmark for absolute performance, the transition toward non-toxic alternatives is increasingly important due to environmental and regulatory considerations, such as restrictions on hazardous substances (RoHS). In this context, our Ag₂Te photodetector provides a viable pathway toward high-performance, environmentally benign MWIR detection. We have incorporated this wavelength-dependent performance comparison into the revised manuscript and Supporting Information.

Initial version:

(line 239) The resulting specific detectivity (D^*) of 1.2×10^9 Jones at 78 K demonstrates competitive performance compared to HgTe photoconductor devices reported in previous reports^{45–48}.

Revised version:

(line 250) The device exhibited a specific detectivity (D^* , 1 kHz) of 1.2×10^9 Jones at 78 K under a bias of 0.02 V, with a measured responsivity of 1.9 mA/W for a device area of 2.6 mm². This D^* value is comparable to those reported for early HgTe CQDs MWIR photodetectors in 2011 (Nature Photonics

5, 489–493), indicating that the present device architecture maintains reliable MWIR detection performance despite employing a non-toxic material system. In addition, we summarize and compare the detectivity of previously reported less-toxic MWIR CQD photodetectors, as well as Ag₂Te CQD-based devices, in Supplementary Table 1. Within this context, the performance of our Ag₂Te photodetector demonstrates a clear advancement among less-toxic MWIR CQD systems reported to date.

Revised version (addition):

Table S1. Performance of the detectivity table of less-toxic MWIR and SWIR Ag₂Te photodetectors from previous studies.

Materials	Detectivity	Wavelength	Year	Ref
InSb	4.7×10 ⁷ Jones	3.0 μm	2025	Nano Lett. 2025, 25, 13549
	3.1×10 ⁷ Jones	3.5 μm		
SnTe	N/A	3.0 μm	2018	ACS Sens. 2018, 3, 10, 2087–2094
Ag₂Se	7.8×10 ⁶ Jones	4.5 μm (Intraband)	2021	ACS Appl. Mater. Interfaces 2021, 13, 49043
Ag₂Te	3.0×10 ¹² Jones	1.3 μm	2024	Nature Photonics volume 18, pages 236–242 (2024)
	9.0×10 ¹⁰ Jones	1.7 μm	2024	ACS Materials Lett. 2024, 6, 4988–4996
	1.0×10 ⁷ Jones	2.0 μm	2024	Adv. Sci. 2024, 11, 2407453
	1.2×10 ⁹ Jones	4.7 μm	2026	This Work

(7) The synthetic approach appears to rely on classical Ostwald ripening processes, yet the authors claim a distinct seeded-growth mechanism without providing mechanistic evidence or theoretical support.

Answer:

We thank Reviewer 1 for the comment. In this manuscript, the term “seeded growth” was used in a descriptive sense to indicate that relatively small Ag₂Te quantum dots are introduced as starting nanocrystals and subsequently evolved into larger particles. We did not intend to imply a strictly seeded-growth mechanism that excludes classical Ostwald ripening processes. However, we acknowledge that the use of the term “seeded growth” may be confusing with a mechanistically distinct seed-mediated growth model. To avoid this ambiguity, we have revised the manuscript and replaced the term “seed Ag₂Te QDs” with “SWIR Ag₂Te QDs”.

Reviewer #2 (Remarks to the Author):

The manuscript by S. Y. Eom et al. reports the synthesis, characterization, and mid-wave infrared (MWIR) photodetector performance of Ag₂Te quantum dots (QDs). The topic is both important and timely, given the increasing market demand for RoHS-compliant infrared semiconductors. Considering that viable semiconductor options become increasingly scarce in the longer-wavelength IR regime, this work is unique and will likely attract broad interest from the Nature Communications readership.

The reported device metrics—including a responsivity of 1.1 A/W, a noise current of 0.26 pA/√Hz, and a resulting specific detectivity of 10⁹ Jones—are attractive. However, additional experimental evidence could further substantiate the claims and strengthen the overall impact of the work.

I recommend publication after the authors address a number of minor and a few major revisions that would help strengthen the discussion and reinforce the conclusions presented in the manuscript.

1. In Figure 1c, the TEM image of the SWIR QD seeds show numerous merged or overlapping “blobs,” with several features appearing larger than 20 nm. If these features are not sintered/fused QDs but are instead particles laying on top of each other, a visible TEM thickness contrast would normally be expected—however, such contrast is not apparent. The major issue is that these larger features are not reflected in the size distribution histogram, which suggests a mismatch between the TEM observations and the quantitative analysis. The authors should clarify the nature of these large features and ensure that the size histogram corresponds to the actual particle size distribution observed in the TEM image.

Answer:

We thank Reviewer 2 for carefully examining the TEM image in Fig. 1c and for raising an important point regarding the apparent presence of significant, merged features in the SWIR Ag₂Te samples.

We agree that several features in Fig. 1c appear significantly larger than the expected nanocrystal size and are not representative of individual, well-isolated quantum dots. Based on repeated TEM observations, we attribute these features primarily to electron-beam-induced damage and coalescence effects, to which the SWIR Ag₂Te nanocrystals are particularly susceptible. Under electron irradiation, the SWIR Ag₂Te nanocrystals frequently undergo rapid morphological changes, including apparent merging or blob-like contrast, which can obscure particle boundaries and complicate thickness-contrast interpretation. As a result, such features do not reliably reflect the intrinsic particle size.

In response to the reviewer’s concern, we emphasize that these prominent features were not treated as individual nanocrystals for size analysis, and we acknowledge that their presence makes quantitative size-distribution extraction from TEM unreliable for the seed samples. Accordingly, we have removed the size-distribution histogram and the associated claims of narrow size dispersion for the SWIR Ag₂Te QDs in the revised manuscript. The TEM images are now used solely for qualitative confirmation of nanocrystal morphology and the relative increase in size after post-growth, rather than for rigorous statistical analysis. We have revised the text to clarify that the reported seed size represents a nominal, representative value, and that beam-induced artifacts limit quantitative size statistics for the SWIR Ag₂Te.

Initial version:

(line 91) TEM analysis demonstrates the increase of nanocrystal size from 6.4 nm to 10.1 nm, which is about 57.8% larger than the seed Ag₂Te.

c

Revised version:

(line 95) TEM analysis reveals a clear increase in nanocrystal size from a nominal SWIR Ag₂Te of ~6.4 nm to ~10.1 nm following post-growth. Due to the pronounced electron-beam sensitivity of SWIR Ag₂Te nanocrystals, TEM images may exhibit merged or blob-like features, which limit reliable extraction of quantitative size distributions.

c

2. In both the main manuscript (k·p approximation) and Supporting Information S3, the authors use a bulk bandgap value of 0.06 eV for monoclinic β -Ag₂Te. This value should be supported with an appropriate literature reference.

Answer:

We thank Reviewer 2 for pointing out the need to properly support the bulk bandgap value used in the k·p approximation. The value of 0.06 eV for monoclinic β -Ag₂Te is taken from well-established experimental literature.

Specifically, Dalven and Gill reported a 0 K bandgap of $E_0=0.064\pm 0.009$ eV for β -Ag₂Te based on temperature-dependent electrical and optical measurements (Phys. Rev. 1966, 143, 666-670). This value has been widely adopted as the reference bulk bandgap for monoclinic β -Ag₂Te in subsequent studies.

In response to the reviewer's comment, we have explicitly added this reference to both the main manuscript (in the k·p approximation discussion) and Supporting Information S3, thereby clarifying the origin of the bulk bandgap parameter used in our analysis.

Initial version:

(SI line 21) A bulk bandgap of 0.06 eV was adopted.

Revised version:

(SI line 21) The bulk bandgap of monoclinic β -Ag₂Te was taken as 0.06 eV, based on the experimentally reported value $E_0 = 0.064\pm 0.009$ eV at 0 K².

3. In manuscript page 7, the terminology EVET should be spelled out first time it is introduced in the manuscript for clarity.

Answer:

We thank Reviewer 2 for pointing this out. In the revised manuscript, EVET is now spelled out as its full name, electronic-to-vibrational energy transfer, at its first occurrence in the main text (line 123), and the abbreviation EVET is used consistently thereafter.

Initial version:

(line 121) EVET

Revised version:

(line 126) electronic-to-vibrational energy transfer (EVET)

4. In Figure 3a, the legend currently labels all curves as being measured “under light.” However, the dotted curves appear to correspond to the dark measurements. The authors should confirm this and correct the legend accordingly.

Additionally, the manuscript should more clearly describe how the responsivity was calculated. Was responsivity determined from the difference between dark and illuminated current at a specific bias? What is the incident optical power? These should be stated to allow readers to reproduce or benchmark the reported figures.

Finally, Figure 3a contains many overlapping curves across different temperatures, which makes it difficult to evaluate the data. To improve clarity, the authors should include raw dark and light I–V curves at each temperature in the Supporting Information, plotted in a linear scale, so readers can clearly examine the underlying device behavior (i.e. magnitudes of dark current vs. photocurrent).

Answer:

We appreciate the reviewer 2’s comments. Regarding the labeling of the J–V curves (298 K and 873 K), we would like to clarify the physical rationale underlying our approach. In conventional NIR photodetector characterization, the state in which a device faces a room-temperature background with no NIR light exposure is often labeled “dark” because such a device typically exhibits negligible photoresponse under ambient thermal conditions.

In contrast, for MWIR detectors, there is a fundamental physical distinction between viewing a room-temperature background and a true dark condition. Owing to the narrow bandgap required for MWIR detection, thermal background radiation at ambient temperature (298 K) can generate a non-negligible photocurrent. Consequently, treating the 298 K condition as “dark” is technically inaccurate for MWIR devices. In this regard, we now provide additional J–V data measured under true dark conditions at 78 K for the same device shown in Fig. 3a.

This measurement demonstrates that the current arises from thermal background with 298 K rather than from the device's intrinsic dark current, thereby validating our labeling and analysis. The measured dark current density is 5.1×10^{-8} A/cm², yielding an $I_{\text{Light}}/I_{\text{Dark}}$ ratio of approximately $\sim 10^3$. This confirms that the photoresponse is clearly distinguishable from the dark current.

Regarding responsivity, we calculated it as the difference between the values under 298 K and 873 K illumination. As suggested, we have included a more detailed description of the responsivity calculation, incident optical power estimation, and J–V curves under true dark conditions in the

Supporting Information due to manuscript length constraints.

Initial version:

(line 197) Fig. 3a shows the results of the J-V curve measurements taken at various operating temperatures, ranging from 78 K to 298 K. The responsivities measured at various temperatures are included in the inset. At a detector temperature of 78 K and under a 0.5 V bias, the responsivity (R) of 1.1 A/W was measured, comparable to the performance of previously reported MWIR HgTe CQDs photodetectors^{36–39}.

Revised version:

(line 204) Fig. 3a shows the J–V characteristics of the MWIR Ag₂Te CQDs on the Pt IDE device, measured at various operating temperatures ranging from 78 K to 298 K. The corresponding linear fits and calculated responsivity method for the temperature-dependent analysis are provided in Supplementary Fig. 10. The corresponding responsivities at various operating temperatures are summarized in the inset. At a detector temperature of 78 K and an applied bias of 0.2 V, a responsivity of 0.3 A/W was achieved. Furthermore, a comparison between the current under true dark conditions and under illumination reveals a difference of $\sim 10^3$. This clear distinction emphasizes the high sensitivity of the Ag₂Te device, even in the presence of thermal background radiation at ambient temperatures.

Revised version (addition):

Figure S10. Linear-scale J–V characteristics of the Ag₂Te CQD photoconductor measured under dark and illuminated conditions at different temperatures. The enlarged view around zero bias shows an approximately linear, symmetric response within ± 0.3 V. At the same time, a slight nonlinearity appears at higher bias, attributed to bulk-limited transport in the CQD film (Nano Lett. 2012, 12, 569–575).

The responsivity was calculated using the following equation:

$$\text{Responsivity} = \left(\frac{I_{\text{Light,873 K}} - I_{\text{Light,298 K}}}{\text{Light source power}} \right)$$

Specifically, the photocurrent is defined as the current difference between 873 K and 298 K

conditions, and the responsivity is calculated using the incident blackbody radiation power at 873 K (28.1 mW/cm²). The Light source power was determined to be 873 K blackbody power. At 0.5 V, the responsivity of 1.1 A/W was measured.

The J–V characteristics were measured under different illumination conditions (dark, 298 K, and 873 K), and the data were replotted on a logarithmic scale to more clearly highlight the differences between the dark and light-induced currents. The measured dark current density and Light current density (873 K) are 5.1×10^{-8} A/cm² is 8.3×10^{-5} A/cm², respectively, yielding an $I_{\text{Light}}/I_{\text{Dark}}$ ratio of $\sim 1.6 \times 10^3$. This confirms that the photoresponse is clearly distinguishable from the dark current.

5. There are few discussions in the manuscript that are confusing to understand:

(1) On page 11 (lines 210–213), the authors state that effective defect reduction may lead to an activation energy larger than the bandgap. It is unclear how this is physically possible. In a semiconductor with negligible defect states, the dominant thermal activation mechanism should be intrinsic carrier generation across the bandgap, which would yield an activation energy equal to the half of the bandgap $E_g/2$ (i.e. how would the activation energy exceed the bandgap)?

Additionally, the manuscript discusses dark-current suppression due to recombination–generation processes occurring “within the depletion region.” It is unclear what depletion region is being referred to, since the device structure is not presented as a diode or heterojunction that would naturally form such a region.

Answer:

We thank Reviewer 2 for the comments and acknowledge that the manuscript's description may have been confusing. The original intention was to emphasize that a larger of the activation energy is indicative of improved device performance. To clarify this point and avoid confusion, we have revised the corresponding text as described below.

Initial version:

(line 210) The activation energy allows the analysis of variations in carrier generation. A larger activation energy compared to the band gap energy infers either effective defect reduction in the quantum dots or suppression of dark current caused by the generation and recombination of thermally generated carriers within the depletion region.

Revised version:

(line 221) The activation energy, 67.9 meV, provides insight into thermally activated carrier generation and transport processes. A larger activation energy infers the reduced defect-assisted conduction and the suppressed dark current from thermally generated carriers, which is beneficial for device performance.

(2) In page 11, hopping frequency equation, the pre-exponent factor ν_0 of 10^{12} /s is used. The manuscript does not explain/discuss how this value was obtained or justified.

Answer:

We appreciate the reviewer 2's comments. To analyze charge transport and dark current behavior in the Ag₂Te CQD film, we adopted the barrier-hopping model proposed by Miller and Abrahams (Phys. Rev. 120, 745). In this framework, the hopping attempt frequency (ν_0) represents the characteristic rate at which charge carriers attempt to overcome potential barriers between adjacent quantum dots (or crystallites).

Physically, this hopping process is phonon-assisted and governed by lattice vibrations. Accordingly, it is common practice to assume that the attempt frequency is on the order of the characteristic phonon frequency of the material system. In our analysis, we employed an attempt frequency of approximately $10^{12}\sim 10^{13}$ Hz (terahertz range, $1\sim 10$ THz = $33\sim 330$ cm⁻¹), which corresponds to a typical lattice vibration timescale widely used in Miller–Abrahams–type hopping models for disordered and nanocrystalline semiconductors.

This assumption is consistent with thermally activated transport, in which charge carriers gain sufficient energy through phonon interactions to hop between localized states. We insert this reference into the hopping-rate calculation.

Initial version:

(line 224) The ν_0 is hopping attempt frequency (phonon frequency). Assuming $\nu_0 = 10^{12} s^{-1}$, the Miller–Abrahams expression gives an average hopping rate $4.6 \times 10^9 s^{-1}$, corresponding to a mean hopping time of $\tau_{hop}(T) = 220 ps$. (Taking $\nu_0 = 10^{13} s^{-1}$ then the $\tau_{hop}(T) = 22 ps$).

Revised version:

(line 236) The average hopping frequency at 78 K was then calculated using the Miller–Abrahams hopping formalism^{40,41}:

$$f_{hop}(T) = \nu_0 \exp \left[-\left(\frac{T_M}{T} \right)^{\frac{1}{4}} \right]$$

where the ν_0 is hopping attempt frequency (phonon frequency). Assuming $\nu_0 = 10^{12} s^{-1}$ based on the reported paper⁴⁰, the Miller–Abrahams expression gives an average hopping rate $4.6 \times 10^9 s^{-1}$, corresponding to a mean hopping time of $\tau_{hop}(T) = 220 ps$. (Taking $\nu_0 = 10^{13} s^{-1}$ then the $\tau_{hop}(T) = 22 ps$).

(3) On page 12 (lines 228 onward), the authors state that the measured hopping frequency is related to rapid photoresponse modulation. Could the authors clarify this connection? Typically, the carrier transit time across the channel—determined by the source–drain separation and the carrier velocity (or related hopping mobility)—is the more relevant parameter governing photoresponse speed.

Also on page 12 (lines 230–231), the authors state, “This can be attributed to the efficient generation of carriers and the rapid discharge under illumination.” Could the authors clarify what is meant by “discharge” in this context? Does this imply a capacitive effect in the device, or is another mechanism responsible for the observed behavior?

It is recommended that the manuscript be revised to address and clarify the issues outlined above (1–3).

Answer:

We appreciate the reviewer 2’s comments. The hopping mechanism describes how it transports charge to another nearby quantum dot. Therefore, the shorter hopping time and the shorter electrode-to-electrode distance result in a faster response time for the device.

The response time is related to device architecture. Some previous studies of SWIR-sensitive Ag₂Te photodetectors demonstrated competitive performance with vertical device architecture. Specifically, we reported a response time of 2.7 μs for ink-processed devices (ACS Materials Letters 2024, 6, 4988–4996). To ensure consistency with these established results and to facilitate a direct performance comparison, we therefore also experiment with a similar vertical structure in the present study.

Finally, we clarify the physical mechanism of the recovery of the photocurrent in the context of our modulation measurements. Upon optical excitation, photogenerated carriers are transported through the CQD film via a series of hopping events. In the previous manuscript, we referred to this recovery process as “carrier discharge,” which may be ambiguous. To avoid confusion, we have revised the text to explicitly describe this process in terms of carrier recombination and extraction from the transport pathways, rather than using the term “discharge.”

Initial version:

(line 230) This can be attributed to the efficient generation of carriers and the rapid discharge under illumination.

Revised version:

(line 243) This can be attributed to the efficient generation and rapid extraction of carriers under illumination.

6. In the time-resolved photoresponse shown in Figures 3d and 4b, the modulation frequency appears too fast relative to the device's response time; specifically, the light is switched off before the device current reaches a stable, steady-state value during the light-on period. While these measurements capture the 'true' photoresponse, the rise and fall time estimates presented in Figure 4b may therefore be inaccurate. It is recommended that the authors include additional data at slower modulation frequencies—where the device current reaches full steady state—for both Figure 3d (lateral device) and Figure 4b (vertical device) in the Supporting Information.

Answer:

We appreciate Reviewer 2's concern regarding the non-ideal modulation waveform and its implications for the device's response time.

First, the photoresponse of the IDE-based photodetector was investigated using a QCL combined with an optical chopper. Because the QCL operates in a pulsed mode, the pulse repetition frequency was fixed at its maximum value of 625 kHz, while the chopper modulation frequency was varied from 20 Hz to 1 kHz. Under pulsed excitation, the measured signal represents the device response to a train of optical impulses rather than to an ideal square-wave modulation. To enable a clearer physical interpretation of the response dynamics, both the step-modulated transient response (Used chopper) and the frequency-dependent normalized photoresponse have been measured.

1. Step-modulated transient response (Used chopper)

Under chopper-modulated illumination at 1000 Hz, we measured a well-defined square-wave photoresponse with a fall time of 124 μs as shown in the figures below and in the Supporting Information. Both rise and fall feature reached the full steady-state regime. The square response is consistently preserved over the frequency range from 100 Hz to 1000 Hz.

(MWIR Ag₂Te on IDE, 78 K, 2900 cm⁻¹, 0.1 V)

At a lower modulation frequency of 20 Hz, although the illumination duration is sufficiently long, the photocurrent does not reach a steady-state plateau. This non-steady-state behavior is attributed to slow current components arising from trap-mediated processes, such as carrier trapping and detrapping. Similar low-frequency transient responses have been widely observed in CQD-based photoconductors and reported in the literature.

For example, in Nature Photonics 19, 1178–1188 (2025), it is noted that intrinsic material effects, such as charge trapping and long RC time constants, can prevent the device from reaching a steady-state plateau. These transient instabilities effectively hinder the immediate stabilization of the current, resulting in the non-flat profiles observed in emerging photodetectors.

2. Frequency-dependent Photoresponse (Direct QCL Modulation)

To further evaluate the device's dynamic bandwidth, frequency-domain measurements were performed by directly tuning the QCL pulse-repetition frequency from 10 Hz to 500 kHz. The photoresponse as a function of modulation frequency is shown below.

From the frequency response, a -3 dB cutoff frequency of 3241 Hz was extracted. This value is in good agreement with the cutoff frequency estimated from the measured fall time using the following relation:

$$\text{Fall-time and bandwidth relationship: } f_{3\text{ dB}} \approx \frac{0.35}{\tau_{fall}} = 2822\text{ Hz } (\tau_{fall}: 124\ \mu\text{s})$$

We confirmed consistency between the time-domain transient response and the frequency-domain bandwidth, and the data were prepared for the manuscript and SI.

Furthermore, to more clearly distinguish the photoresponse duration and enhance the reliability of response-speed evaluation in the vertical device configuration, we measured the pulse-frequency-dependent photoresponse intensity. This approach enables a robust, quantitative characterization of the photodetector's temporal behavior and operational bandwidth.

Accordingly, we have extensively revised the contents of Figures 3 and 4 to provide a more comprehensive analysis.

Initial version:

(line 228) These findings confirm the ability for rapid modulation, as shown in Fig. 3d. Clear on-off responses were observed at modulation frequencies from 100 Hz to 1 kHz under a 600 °C blackbody radiation. This can be attributed to the efficient generation and rapid extraction of carriers under illumination.

(line 279) Under photoexcitation by a quantum cascade laser at 2800 cm⁻¹(3.6 μm), modulated at 625 kHz, the as-fabricated vertical-type Ag₂Te photodetector exhibited a nanosecond-scale photoresponse, with a rise time of 230 ns and a fall time of 576 ns. Such measured response times show competitive performance compared to previous reports.

Revised version:

Fig. 3| Device performance of MWIR Ag₂Te photodetectors. (Interdigitated electrode, IDE) a, J–V characteristics measured at various temperatures. (inset: responsivity versus operating temperature). **b,** Activation energy extracted from temperature-dependent measurements. **c,** Conductance G as a function of T^{-1/4}. **d,** Temporal photoresponse at 1 kHz under 2900 cm⁻¹ laser by Quantum Cascade Laser (QCL). **e,** The noise current density recorded under dark conditions. **f,** Time-dependent current curve at 0.1 V derived from blackbody radiation at 20, 37, 40 °C. **g,** Schematic structure of the MWIR Ag₂Te device integrated on the IDE. **h,** Infrared thermal imaging setup and corresponding images confirm the photoresponse of the MWIR Ag₂Te device.

(line 240) These findings confirm the ability for rapid modulation, as shown in Fig. 3d and Supplementary Fig. 12. Under illumination from the QCL at 2900 cm⁻¹, the device exhibited clear on–off photoresponses at a modulation frequency of 1 kHz, with a rise time of 141 μs and a fall time of 124 μs. This can be attributed to the efficient generation and the rapid extraction of carriers under illumination.

Fig. 4| Device performance of MWIR Ag₂Te photodetectors. (Vertical structure) a, Schematic of MWIR Ag₂Te photoconductor device and a cross-sectional focused ion beam (FIB) image **b**, On-off characteristics with a 2900 cm⁻¹ mid-infrared QCL. The pulse duration was 625 kHz. **c**, The response time of the MWIR Ag₂Te QD photodetector plotted with those of a commercial MCT bulk crystal photodetector (Blue, Kolmar Tech., KMPV11-1-J1) and a commercial InSb bulk crystal photodetector (Grey, Infrared Associates, 1176-2C-14-0.5). **d**, Time-domain photocurrent responses measured under QCL pulse excitation with repetition frequencies ranging from 5 Hz to 500 kHz. **e**, Corresponding normalized photoresponse as a function of modulation frequency, exhibiting a -3 dB cutoff frequency of 69.6 kHz.

(line 301) The vertical device exhibits a measurable photoresponse under blackbody illumination at 873 K, indicating effective infrared photodetection (Supplementary Fig. 14). Under photoexcitation by a QCL at 2900 cm⁻¹ (3.4 μm), modulated at 200 kHz, the as-fabricated vertical-type Ag₂Te photodetector exhibited a nanosecond-scale photoresponse, with a rise time of 211 ns and a fall time of 523 ns. Such measured response times show competitive performance compared to previous reports.

(line 317) To assess the robustness of the device response under repetitive optical excitation, pulse-frequency-dependent photoresponse measurements were performed over a modulation range from 5 Hz to 500 kHz (Fig. 4d). The photocurrent amplitude gradually decreased with increasing repetition frequency, reflecting the device's finite response dynamics. Fig. 3e shows the normalized frequency response yields a -3 dB cutoff frequency of 69.6 kHz. This frequency-domain analysis provides independent validation of the high-speed temporal response, consistent with the trends observed in transient pulse measurements.

Revised version (addition):

Figure S12. Temporal and frequency-dependent photoresponse of the device. (a) Time-domain photoresponse measured under optical chopper modulation, showing distinct on–off switching behavior. (b) Pulse-frequency-dependent photoresponse measured by directly modulating the QCL pulse repetition frequency, illustrating the evolution of the response amplitude as a function of pulse frequency.

From the normalized frequency response, a -3 dB cutoff frequency of 3.2 kHz was extracted. This value is in good agreement with the cutoff frequency estimated from the measured fall time using the following relation:

$$\text{Fall-time and bandwidth relationship: } f_{3\text{ dB}} \approx \frac{0.35}{\tau_{\text{fall}}} = 2.8\text{ Hz} \quad (\tau_{\text{fall}}: 124\ \mu\text{s})$$

The bandwidth derived from the pulse measurements is in good agreement with the data shown in Fig. 3d.

Figure S14. (a) Current density–voltage (J-V) characteristics of the device measured in the dark and under illumination. The net photocurrent is shown, calculated as the difference between the two curves. (b) On–off photoresponse was measured using an optical chopper, while the intrinsic response time was evaluated using a pulsed laser because the chopper’s modulation frequency was limited to 1 kHz. (c) Responsivity was calculated from current and power densities. The device’s active area is 1.05 mm². At 1 V (positive bias), the responsivity is approximately 40 mA/W. The lower responsivity relative to the IDE structure is attributed to the ITO electrode’s low infrared transmittance.

7. The current noise data shown in Figure 3e require a more detailed analysis. At present, the manuscript only reports the noise value at 1 kHz. Could the authors clarify the dominant source of noise? For assembled QD films, this is typically 1/f noise. Additionally, is the 1/f knee around 100 Hz in Figure 3e accurate, or could the measured noise level have reached the instrument noise floor (approximately 0.5 pA/ $\sqrt{\text{Hz}}$ for this type of setup) starting near 100 Hz? Providing a more thorough discussion or additional measurements would strengthen the presented data.

Answer:

We appreciate the reviewer 2's comments. In this work, the responsivity used to calculate the detectivity was determined from photocurrent modulation at 1 kHz. The detectivity was evaluated using the noise current measured at the same frequency, ensuring a self-consistent definition of D^* under the device's actual operating conditions.

As noted by the reviewer's comment, the noise spectrum reaches a near-constant noise floor around ~100-200 Hz, and using the noise value in this frequency range would also be acceptable. However, it is common practice in the photodetector literature to evaluate detectivity at higher modulation frequencies, such as 1 kHz or 10 kHz, where the noise spectrum is flat and less affected by low-frequency flicker noise (e.g., Nature Communications 10, 2125 (2019); Advanced Science 7, 2000068 (2020)); Advanced Science 11, 2407453 (2024)) In line with this convention, and because our responsivity was measured at a 1 kHz chopping frequency, we used the 1 kHz noise value to calculate D^* (1 kHz). This clarification has been added to the manuscript.

In addition, as suggested by the reviewer, we consider the possibility that the measured noise could approach the instrumental noise at low frequencies. To address this point, we have included additional data in the Supporting Information comparing the device noise with the background noise of the measurement system. The instrument noise floor was measured by shorting the input of the current preamplifier to ground and was found to be significantly lower than the noise measured with the device. The results confirm that the reported noise values are dominated by the device rather than the instrumentation.

Based on the dark current and the dynamic resistance measured at a bias of 0.02 V, the theoretical noise contributions were estimated.

The shot-noise-limited current noise: $0.23 \text{ pA/Hz}^{0.5}$, $i_{shot} = \sqrt{2qI\Delta f}$

The Johnson–Nyquist thermal noise: $0.19 \text{ pA/Hz}^{0.5}$, $i_{theraml} = \sqrt{4k_B R \Delta f}$

The experimentally measured noise density of $0.26 \text{ pA/Hz}^{0.5}$ at 1000 Hz shows close agreement with these theoretical limits. This correspondence indicates that at a modulation frequency of 1000 Hz, low-frequency $1/f$ noise is effectively suppressed and the device operates near its intrinsic noise floor.

Accordingly, we used the measured noise density of $0.26 \text{ pA/Hz}^{0.5}$ for the detectivity calculation, as it provides a more realistic and physically justified assessment of the device performance under operational conditions. The manuscript has been revised to clarify this analysis.

Based on these considerations, we have revised the manuscript accordingly and added the corresponding background-noise data to the Supporting Information.

Initial version:

(line 233) Under cryogenic conditions, the noise level was measured to be $0.26 \text{ pA/Hz}^{0.5}$ at 1000 Hz. Using this value, the specific detectivity was calculated with the following equation,

$$D^* = \frac{R\sqrt{A}}{i_n}$$

where R , A , and i_n are the responsivity, device area, and noise current density, respectively.

The resulting specific detectivity (D^*) of 1.2×10^9 Jones at 78 K demonstrates competitive performance compared to HgTe photoconductor devices reported in previous reports⁴⁵⁻⁴⁸

Revised version:

(line 244) Under cryogenic conditions, the noise level was measured to be $0.26 \text{ pA}/\sqrt{\text{Hz}}$ at 1 kHz, as detailed in Supplementary Fig. 13. Using this value, the specific detectivity was calculated with the following equation,

$$D^* = \frac{R\sqrt{A}}{i_n}$$

where R , A , and i_n are the responsivity, device area, and noise current density, respectively.

The device exhibited a specific detectivity (D^* , 1 kHz) of 1.2×10^9 Jones at 78 K under a bias of 0.02 V, with a measured responsivity of 1.9 mA/W for a device area of 2.6 mm^2 . This D^* value is comparable to those reported for early HgTe CQDs MWIR photodetectors in 2011⁴², indicating that the present device architecture maintains reliable MWIR detection performance. In addition, we summarize and compare the detectivity of previously reported non-toxic MWIR CQD photodetectors, as well as the Ag_2Te CQD photodetector, in Supplementary Table 1.

Revised version (addition):

Figure S13. The noise spectral density of the Ag_2Te photodetector was measured under dark conditions. At low frequencies (below ~ 100 Hz), the noise exhibits a characteristic $1/f$ dependence. As the frequency increases, the noise spectrum transitions to a frequency-independent region above ~ 100 Hz, corresponding to a white-noise-dominated regime.

Based on the dark current and the dynamic resistance measured at a bias of 0.02 V, the theoretical noise contributions were estimated.

The shot-noise-limited current noise: $0.23 \text{ pA}\sqrt{\text{Hz}}$, $i_{\text{shot}} = \sqrt{2qI\Delta f}$

The Johnson–Nyquist thermal noise: $0.19 \text{ pA}\sqrt{\text{Hz}}$, $i_{\text{thermal}} = \sqrt{4k_B R \Delta f}$

The experimentally measured noise density of $0.26 \text{ pA}/\text{Hz}^{0.5}$ at 1 kHz shows close agreement with these theoretical limits. This correspondence indicates that at a modulation frequency of 1 kHz, low-frequency $1/f$ noise is effectively suppressed and the device operates near its intrinsic noise floor.

For comparison, the instrument noise floor of the measurement system was independently measured by shorting the input of the current preamplifier to ground. The resulting background noise level is significantly lower than the noise measured with the device across the entire frequency range of interest, including at 1 kHz. This confirms that the noise used for detectivity evaluation is dominated by the intrinsic device noise rather than by the measurement electronics.

8. On page 14 (lines 272–273), the authors state that an atomic ligand was used. The specific type of ligand should be clearly specified in the manuscript to ensure reproducibility and clarity.

Answer:

We thank Reviewer 2 for pointing out the ambiguity regarding the atomic ligand. In the revised manuscript, we have explicitly specified the type of atomic ligand used to ensure clarity and reproducibility. Specifically, the term “atomic ligand” has been replaced with cetyltrimethylammonium bromide (CTAB), and the corresponding ligand treatment procedure is now clearly referenced in the Method section.

Initial version:

(line 277) Atomic ligand

Revised version:

(line 296) A cetyltrimethylammonium bromide (CTAB)

9. Just out of curiosity regarding the Ag₂Te QDs (not requiring a mandatory revision): Some metal chalcogenide QDs, such as HgSe and Ag₂Se, begin to exhibit intraband characteristics as their size increases. Could the current MWIR Ag₂Te QDs presented in this work potentially display intraband behavior as well?

Answer:

We thank Reviewer 2's comment. As the reviewer suggests, intraband absorption has been observed in several narrow bandgap metal chalcogenide quantum dot systems, such as HgSe and Ag₂Se. However, at this moment, it is somewhat difficult to predict the possibility. While the current work does not provide direct experimental evidence for intraband transitions, we agree that the size and compositional regime explored here lie near the boundary at which intraband transition may begin to emerge. A systematic investigation of self-doping, electrostatic gating, or further size enlargement would be required to probe this regime unambiguously. Such studies represent an interesting direction for future work beyond the scope of the present manuscript. I would like to thank Reviewer 2 for asking this question.

Reviewer #3 (Remarks to the Author):

In this manuscript, Eom et al. reported a synthetic method for large Ag₂Te QDs with absorption edge extending into MWIR regime, and demonstrated photoconductive detectors exhibiting photoresponse under MWIR illumination. The extension of Ag₂Te QD absorption into the MWIR is potentially interesting. However, in its current form, the manuscript does not provide sufficient mechanistic insights into the growth process, nor does it convincingly advance the understanding of device physics. The work is therefore more of an incremental materials/demonstration paper and of preliminary nature rather than a comprehensive study suitable for Nature Communications.

Below are specific comments:

1. In Fig. 1c, the TEM images of the seed QDs showed clear agglomeration, whereas in Fig. 1f the larger QDs appeared well separated and more uniformly dispersed. The authors should provide a clear explanation for this discrepancy. Is it due to differences in surface chemistry, ligand coverage, sample preparation for TEM, or actual differences in colloidal stability between seeds and grown QDs?

Answer:

We thank the reviewer for this careful comment on the difference in dispersion between the SWIR Ag₂Te QDs in Fig. 1c and the post-growth larger QDs in Fig. 1f.

The observed discrepancy arises from differences in surface chemistry and electron-beam sensitivity, rather than from inconsistencies in the synthesis or analysis. The seed-stage Ag₂Te nanocrystal are particularly sensitive to electron-beam irradiation with 200 kV of acceleration voltage, and during TEM imaging, they frequently undergo beam-induced morphological changes, including apparent agglomeration or blob-like contrast. As discussed in the revised manuscript, such beam-induced effects obscure particle boundaries and can give the appearance of agglomeration even when the nanocrystals are well dispersed in solution before deposition.

In contrast, the larger Ag₂Te QDs obtained after post-growth exhibit improved structural robustness and enhanced surface passivation, owing to increased nanocrystal size and modified ligand coverage. These factors significantly reduce electron-beam-induced coalescence during TEM imaging, resulting in images with more clearly separated, uniformly dispersed particles.

We emphasize that this difference should not be interpreted as a fundamental change in colloidal stability between the two samples. Instead, it reflects the greater susceptibility of smaller, seed-stage nanocrystals to beam-induced artifacts during TEM characterization, combined with the improved stability of the post-grown QDs under identical imaging conditions. To avoid misinterpretation, we have revised the manuscript to clarify this point and to limit the use of TEM images of seed-stage QDs a qualitative morphology assessment.

Initial version:

(line 91) TEM analysis demonstrates the increase of nanocrystal size from 6.4 nm to 10.1 nm, which is about 57.8% larger than the seed Ag₂Te.

c

Revised version:

(line 92) TEM analysis reveals a clear increase in nanocrystal size from a nominal SWIR Ag₂Te of ~6.4 nm to ~10.1 nm following post-growth (~57.8% increase). Due to the pronounced electron-beam sensitivity of SWIR Ag₂Te nanocrystals, TEM images may exhibit merged or blob-like features, which limit reliable extraction of quantitative size distributions.

c

2. The TEM image of the seeds is not shown at an appropriate magnification to reliably determine the average size. A low-magnification image is required to assess the overall morphology. While the seed absorption spectra show distinct transitions, the larger dots do not exhibit clear excitonic features despite appearing size-uniform in TEM. The authors should provide a clear explanation for this discrepancy.

Answer:

We thank Reviewer 3 for this thoughtful comment, which addresses both the TEM characterization and the optical response of the Ag₂Te quantum dots.

We agree that the TEM image of the seed-stage Ag₂Te QDs shown in Fig. 1c is not optimized for reliable average-size measurements, and that a lower magnification image would be more appropriate for assessing overall morphology. As discussed in the revised manuscript, however, quantitative size analysis of the seed-stage QDs by TEM is intrinsically limited by their pronounced electron-beam sensitivity, which leads to beam-induced morphological changes. For this reason, TEM images of the seeds are not intended for rigorous size determination and are used solely for qualitative morphology assessment.

Regarding the optical response, we acknowledge the reviewer's observation that the seed absorption spectra exhibit more distinct transition. In contrast, the larger post-grown QDs do not show well-resolved excitonic features despite appearing relatively uniform in TEM. This apparent discrepancy reflects a fundamental difference between structural uniformity and spectroscopic linewidth in narrow bandgap quantum dots.

As the Ag₂Te nanocrystals grow larger and approach the weakly confined regime, several effects contribute to the loss of sharp excitonic features:

- (i) Reduced quantum confinement, because the diameter of the MWIR Ag₂Te CQDs (~10 nm) is comparable to the exciton Bohr radius of Ag₂Te (~5 nm) (Andrey L. Rogach *et al.*, *Adv. Optical Mater.* **2025**, *13*, 2403489), placing the system in the weak confinement regime where electronic states become closely spaced and excitonic features are strongly broadened.
- (ii) Increased electronic states broadening arising from size dispersion at the ensemble level, compositional fluctuations, and surface disorder.
- (iii) The emergence of band-tail states extending in the MWIR, which smear the absorption onset even when the size of the particles is uniform in TEM.

Consequently, the absence of well-defined excitonic peaks in the larger Ag₂Te QDs does not contradict their apparent uniformity in TEM, but rather reflects the intrinsic broadening expected for narrow bandgap, weakly confined nanocrystals in the MWIR regime. We have clarified this point in the revised manuscript to distinguish between structural observations from TEM and ensemble-averaged optical responses.

Initial version:

(line 90) TEM analysis demonstrates the increase of nanocrystal size from 6.4 nm to 10.1 nm, which is about 57.8% larger than the seed Ag₂Te. Supplementary Figs. 1-2 show the TEM image of QDs synthesized, confirming a spherical morphology and a narrow size distribution (standard deviation < 12%). To confirm the size-dependent bandgap energy, we employed a two-band k·p approximation.

(Supplementary Fig. 3).

Revised version:

(line 95) TEM analysis reveals a clear increase in nanocrystal size from a nominal SWIR Ag₂Te of ~6.4 nm to ~10.1 nm following post-growth (~57.8% increase). Due to the pronounced electron-beam sensitivity of SWIR Ag₂Te nanocrystals, TEM images may exhibit merged or blob-like features, which limit reliable extraction of quantitative size distributions. Supplementary Figs. 1 and 2 present representative TEM image of Ag₂Te QDs synthesized with varying post-growth reaction times, showing a spherical morphology and clear size evolution with prolonged growth. The associated absorption spectra reveal a consistent red-shift with increasing CQD size. The resulting size-dependent bandgap trend is further analyzed using a two-band k·p approximation, as shown in Supplementary Fig. 3.

3. The role of the reducing agents is not clearly elucidated. The authors stated that oleylamine can reduce Ag^+ to Ag nanoparticles, yet they intentionally introduced an additional reducing agent. This raises the concern that metallic Ag nanoparticles could form during synthesis. The authors should explicitly explain whether Ag nanoparticles are indeed formed under the reported conditions and why an extra reducing agent is necessary if oleylamine already has reducing capability, and how its concentration affects QDs' phase purity and sizes.

Answer:

We thank the reviewer for raising this important point regarding the role of the reducing agents and the potential formation of metallic Ag nanoparticles during synthesis. We agree that oleylamine may act as a mild reducing agent for Ag^+ under certain conditions. However, in the present synthesis, oleylamine primarily functions as a coordinating and surface-passivating ligand rather than an effective reducing agent, due to the limited thermodynamic driving force and slow reducing kinetics for $\text{Ag}^+ \rightarrow \text{Ag}(0)$ under the conditions (The standard reduction potential (E°) of Ag^+/Ag at 298 K is 0.8 V)

The surface passivation of long organic ligand (oleylamine) that strongly attaches to Ag^+ ions can act as an energy barrier to introduce new Ag^+ ions into the SWIR Ag_2Te QDs, making it challenging to enlarge them. Hence, we added additional reducing agent to decrease this energy barrier. The role of additional reducing agent is to weaken the bond between the Ag^+ ions and oleylamine on the surface of the QDs to ensure that the supplied silver is efficiently incorporated into the Ag_2Te lattice. This eventually facilitates the post-growth. In the absence of a more potent reducing agent, post-growth process is slow and spatially inhomogeneous, which leads to incomplete growth or uncontrolled side reactions.

Importantly, we do not observe evidence for the formation of metallic Ag nanoparticles under the reported conditions. TEM imaging reveals no high-contrast metallic Ag domains, and no distinct plasmonic features characteristic of Ag nanoparticles are observed in the optical spectra. Instead, the post-growth products consistently exhibit larger size Ag_2Te nanocrystals with preserved semiconductor-like optical behavior, indicating that Ag incorporation proceeds via lattice growth rather than phase separation.

Regarding concentration effects, we note that the reducing agent concentration was deliberately kept low and within a narrow window optimized for phase-pure growth. Excessive concentrations of reducing agents promoted uncontrolled reduction and degraded product quality, whereas insufficient amounts led to incomplete growth. These observations are consistent with a reduction-rate-controlled growth regime, rather than metallic Ag nanoparticle formation. To avoid overinterpretation, we have clarified this role qualitatively in the revised manuscript without claiming a detailed kinetic model.

Initial version:

(line 439) **Synthesis of MWIR Ag_2Te**

The 10 mg/mL SWIR Ag_2Te solution (in oleylamine) was stirred under vacuum (<100 mTorr) for 1 h. After degassing, increase the reaction temperature to 180°C , then rapidly inject 0.1 M silver nitrate (in oleylamine) and the reducing agent (Diphenylphosphine or Alane N,N-dimethylethylamine), and allow the reaction to proceed for 2–6 h.

Revised version:

(line 455) **Synthesis of MWIR Ag₂Te**

Pre-formed SWIR Ag₂Te nanocrystals dispersed in OLAm (10 mg/mL) were used as the starting material. The nanocrystal solution was degassed under vacuum (<100 mTorr) for 1 h and subsequently heated to 180 °C under an inert atmosphere. Then, 0.1 mL of 0.1 M AgNO₃ solution in OLAm and 0.02 mL of reducing agent (DPP, Alane N,N-dimethylethylamine) were rapidly injected to initiate post-growth. An external reducing agent was added to weaken the bond between Ag⁺ ions and oleylamine on the surface of nanocrystals to lower the surface energy barrier and introduce additional silver. The reaction was maintained at 180 °C for 2-6 h, during which the nanocrystal size was tuned by varying the post-growth duration. Longer reaction times resulted in larger Ag₂Te nanocrystals and a corresponding red-shift of the absorption edge into the MWIR region (Supplementary Figs. 1-2). After completion, the reaction mixture was cooled to room temperature, and the products were purified by repeated precipitation and redispersion for further characterization and device fabrication.

4. During synthesis, the authors used reducing agents at high temperature, which can readily reduce Ag^+ to metallic $\text{Ag}(0)$. This may significantly increase the metallic Ag content in the QDs. However, the manuscript does not adequately discuss the presence of $\text{Ag}(0)$ or its influence on QD growth and the final product. The authors cite only one reference related to PbSe synthesis, which is not directly relevant considering the different reduction/oxidation potential of Ag and Pb. I suggest including the Ag 3d XPS spectra of the seed QDs together with those of the MWIR Ag_2Te QDs. This comparison would be more informative, as the seed Ag_2Te synthesis does not require a reducing agent, whereas the MWIR Ag_2Te QDs do. Additionally, XRD alone cannot confirm or exclude the presence of a certain fraction of metallic $\text{Ag}(0)$ within the QDs.

Answer:

We thank Reviewer 3 for the comment. We would first like to clarify a potential misunderstanding regarding the XPS data. The Ag 3d XPS spectra, included in the previous manuscript, correspond to the MWIR Ag_2Te QDs synthesized under the post-growth method, not the SWIR Ag_2Te QDs. These data are added to directly address the concern that the use of reducing agents at elevated temperatures could lead to the formation of metallic $\text{Ag}(0)$ in MWIR Ag_2Te QDs, where the formation of metallic $\text{Ag}(0)$ would be most likely to occur.

To avoid any ambiguity and to fully address the reviewer's suggestion, we have now included Ag and Te XPS spectra of both the SWIR Ag_2Te QDs and the MWIR Ag_2Te QDs in the revised Supporting Information (Supplementary Fig. 4). This direct comparison is particularly informative to compare the two. Importantly, the Ag $3d_{5/2}$ and Ag $3d_{3/2}$ core-level spectra of the SWIR and MWIR Ag_2Te CQDs are highly similar, with peak positions characteristic of Ag^+ in Ag-Te bonding environments. There is no discernible difference between the two samples within the detection limits. In particular, no additional features or pronounced shoulders near ~ 368.5 eV or ~ 374.5 eV—commonly associated with metallic $\text{Ag}(0)$ —are observed in the MWIR Ag_2Te relative to the SWIR reference. The close correspondence between the SWIR and MWIR Ag 3d spectra demonstrates that the introduction of a reducing agent is not used for the formation of metallic Ag nanoparticles, but instead facilitates controlled incorporation of Ag into the Ag_2Te lattice during post-growth. Consistent with this conclusion, no plasmonic absorption features associated with metallic Ag nanoparticles are observed in the optical spectra.

Initial version:

Figure S4. XPS spectra of Ag 3d and Te 3d core levels in the Ag₂Te CQDs. XPS analysis confirmed that the seed Ag₂Te composition was maintained, with clear Ag and Te peaks observed. The Ag 3d region showed a 3d_{3/2} peak at 373.8 eV and a 3d_{5/2} peak at 368.2 eV, while the Te 3d region displayed a 3d_{3/2} peak at 582.6 eV and a 3d_{5/2} peak at 572.0 eV. It is worth noting that no residual metallic Ag(0) peaks appeared in the XPS spectra, consistent with the absence of Ag (111) or (200) peaks in the XRD pattern. For Te, the Te–O peak at 574.4 eV was only slightly observed, indicating that Ag₂Te CQDs possess inherent stability against surface oxidation, which is beneficial for maintaining their electronic properties during device operation.

Revised version:

Figure S4. XPS spectra a. Ag 3d and Te 3d core levels XPS spectra of SWIR Ag₂Te CQDs

b. Ag 3d and Te 3d core levels XPS spectra of MWIR Ag₂Te CQDs synthesized via post-growth method. XPS analysis shows that the Ag₂Te CQD composition is maintained when the CQDs further grow from SWIR CQD ($2r= 6.3$ nm, **a**) to the MWIR CQD ($2r= 10.4$ nm, **b**). The Ag 3d_{3/2}, Ag 3d_{5/2}, Te 3d_{3/2}, Te 3d_{5/2} appear at 373.8 eV, 368.2 eV, 582.6 eV, and 572.0 eV, respectively. No discernible Ag(0) or Te-O is identified in the XPS spectra.

5. The Gaussian peak fitting presented in Fig. 1e is not convincing. The experimental absorption spectra didn't show a well-defined excitonic peak; rather, they exhibit a broad shoulder and likely to be a tail extending into the MWIR. Moreover, the fitted curve does not overlap with the long-wavelength edge particularly well. The Gaussian fitting presented appears somewhat arbitrary and if one plots the absorption spectra together in the same plot they will overlap to a large extent especially towards lower energies. Moreover there appears to be significant discrepancies between TEM images and size histograms in supporting Fig 1 and 2.

Answer:

We thank the reviewer for the detailed review. We agree that the experimental absorption spectra do not exhibit well-defined excitonic peaks, but instead show a broad shoulder with a long tail extending into the MWIR.

In response, we have removed the Gaussian fit function from Fig. 1e and eliminated the associated “band-edge extraction” discussion in the revised manuscript. The absorption spectra are now presented without a fit function, and the manuscript has been revised to avoid overinterpretation of these broad spectral features.

We also acknowledge the reviewer's concerns regarding discrepancies between the TEM image and the size histograms in Supplementary Figs. 1 and 2.

As clarified in the revised manuscript, quantitative TEM size statistics for the pre-formed (SWIR) Ag₂Te nanocrystals are limited by the electron-beam sensitivity, which can induce apparent coalescence. Accordingly, we have removed the TEM size-distribution histograms and any claims of narrow size dispersion, and the TEM data are now used only for qualitative assessment of morphology and size evolution.

Initial version:

(link 89) The Ag₂Te, produced through the seeded growth process, exhibits absorption at 3220 cm⁻¹ (FWHM 1650 cm⁻¹), indicating mid-IR absorption (Fig. 1e).

(line 91) TEM analysis demonstrates the increase of nanocrystal size from 6.4 nm to 10.1 nm, which

is about 57.8% larger than the seed Ag_2Te .

c

Revised version:

(line 94) The Ag_2Te CQDs obtained through the post-growth process exhibit broadband absorption extending into 5 μm , the MWIR region, confirming MWIR optical response (Fig. 1e).

(line 95) TEM analysis reveals a clear increase in nanocrystal size from a nominal SWIR Ag_2Te of ~ 6.4 nm to ~ 10.1 nm following post-growth ($\sim 57.8\%$ increase). Due to the electron-beam sensitivity of SWIR Ag_2Te nanocrystals, TEM images may exhibit merged or blob-like features, which limit reliable extraction of quantitative size distributions.

c

6. In the activation energy analysis, it is not clear how the resistance values were extracted from the JV characteristics. The manuscript should specify the exact procedure used to obtain resistance from JV curves (e.g., from the low-bias slope, linear fits, or another method?). Furthermore, since the devices are photoconductors, it would be more appropriate to present JV curves on a linear scale to assess linearity and possible non-ohmic effects. More importantly, the device structure was not clearly described in the main text. A detailed description of the device stack and geometry must be provided.

Answer

We appreciate the reviewer's comment on the extraction of the activation energy and the interpretation of the J–V characteristics. The resistance values used for calculating the activation energy were obtained by $R_{diff} = \left(\frac{dV}{dJ}\right)$.

For clarity, we have added the corresponding J–V curves to illustrate the transport behavior. As shown in the linear-scale J–V characteristics, the device exhibits an approximately linear response over the bias range -0.3 to 0.3 V. At higher applied voltages, a gradual change in slope is observed. Such behavior is commonly reported in quantum-dot-based photoconductive devices. It is attributed to bulk transport effects, including field-assisted hopping, modifications of the effective hopping barrier, and trap-related processes under higher electric fields, rather than contact-limited behavior (Nano Lett. 2012, 12, 569–575).

We have revised the activation energy method in the manuscript, and the J–V data and additional discussion are inserted in the Supporting Information.

Initial version:

(line 197) Fig. 3a shows the results of the J–V curve measurements taken at various operating temperatures, ranging from 78 K to 298 K.

(line 208) where $R_0A(T)$, $R_0A(0)$, E_a , k_B , T the temperature-dependent resistance–area product, the resistance–area product at 0 K, activation energy, Boltzmann's constant, temperature, respectively.

Revised version:

(line 204) Fig. 3a shows the J–V characteristics of the MWIR Ag₂Te CQDs on the Pt IDE device, measured at various operating temperatures ranging from 78 K to 298 K. The corresponding linear fits and the calculated responsivity method for the temperature-dependent analysis are provided in Supplementary Fig. 10.

(line 218) where $R_0A(T)$, $R_0A(0)$, E_a , k_B , T the temperature-dependent resistance–area product from $R_{diff} = \left(\frac{dV}{dJ}\right)$ at 0.5 V, the resistance–area product at 0 V, activation energy, Boltzmann’s constant, temperature, respectively.

Revised version (addition):

Figure S10. Linear-scale J–V characteristics of the Ag₂Te CQD photoconductor measured under dark and illuminated conditions at different temperatures. The enlarged view around zero bias shows an approximately linear, symmetric response within ± 0.3 V. At the same time, a slight nonlinearity appears at higher bias, attributed to bulk-limited transport in the CQD film⁷.

The responsivity was calculated using the following equation:

$$\text{Responsivity} = \left(\frac{I_{\text{Light,873 K}} - I_{\text{Light,298 K}}}{\text{Light source power}} \right)$$

Specifically, the photocurrent is defined as the current difference between 873 K and 298 K conditions, and the responsivity is calculated using the incident blackbody radiation power at 873 K (28.1 mW/cm^2). The light source power was determined to be 873 K blackbody power. At 0.5 V, the responsivity of 1.1 A/W was measured.

The J–V characteristics were measured under different illumination conditions (dark, 298 K, and 873 K), and the data were replotted on a logarithmic scale to more clearly highlight the differences between the dark and light-induced currents. The measured dark current density and the light current density (873 K) are $5.1 \times 10^{-8} \text{ A/cm}^2$ and $8.3 \times 10^{-5} \text{ A/cm}^2$, respectively, yielding an $I_{\text{Light}}/I_{\text{Dark}}$ ratio of $\sim 1.6 \times 10^3$. This confirms that the photoresponse is clearly distinguishable from the dark current.

7. L224, the authors assigned a value for the hopping frequency without any justification. Since this parameter directly influences the hopping rate and response time, its choice cannot be arbitrary. The authors should justify their choices.

Answer:

We appreciate the reviewer's comments. To analyze charge transport and dark current behavior in the Ag₂Te CQD film, we adopted the barrier-hopping model proposed by Miller and Abrahams (Phys. Rev. 120, 745). In this framework, the hopping attempt frequency (ν_0) represents the characteristic rate at which charge carriers attempt to overcome potential barriers between adjacent quantum dots (or crystallites).

Physically, this hopping process is phonon-assisted. It is common practice to assume that the attempt frequency is on the order of the characteristic phonon frequency of the material system. In our analysis, we employed an attempt frequency of approximately $10^{12}\sim 10^{13}$ Hz (terahertz range, $1\sim 10$ THz = $33\sim 330$ cm⁻¹), which corresponds to a typical lattice vibration timescale widely used in Miller–Abrahams–type hopping models for disordered and nanocrystalline semiconductors.

This assumption is consistent with thermally activated transport, in which charge carriers gain sufficient energy through phonon interactions to hop between localized states. We insert this reference into the hopping-rate calculation.

Initial version:

(line 224) The ν_0 is hopping attempt frequency (phonon frequency). Assuming $\nu_0 = 10^{12} \text{ s}^{-1}$, the Miller–Abrahams expression gives an average hopping rate $4.6 \times 10^9 \text{ s}^{-1}$, corresponding to a mean hopping time of $\tau_{hop}(T) = 220 \text{ ps}$. (Taking $\nu_0 = 10^{13} \text{ s}^{-1}$ then the $\tau_{hop}(T) = 22 \text{ ps}$).

Revised version:

(line 233) The average hopping frequency at 78 K was then calculated using the Miller–Abrahams hopping formalism^{40,41}:

$$f_{hop}(T) = \nu_0 \exp \left[-\left(\frac{T_M}{T} \right)^{\frac{1}{4}} \right]$$

where the ν_0 is hopping attempt frequency (phonon frequency). Assuming $\nu_0 = 10^{12} \text{ s}^{-1}$ based on the reported paper⁴⁰, the Miller–Abrahams expression gives an average hopping rate $4.6 \times 10^9 \text{ s}^{-1}$, corresponding to a mean hopping time of $\tau_{hop}(T) = 220 \text{ ps}$. (Taking $\nu_0 = 10^{13} \text{ s}^{-1}$ then the $\tau_{hop}(T) = 22 \text{ ps}$).

==

8. L239, the authors claim that their device performance is “comparable” to HgTe QD devices. However, the reported specific detectivity appears to be 2 orders of magnitude lower than state-of-the-art HgTe QD detectors (10^9 vs 10^{11} Jones, see: *Nat. Photon.* 18, 1147–1154 (2024); *Adv. Mater.* 2025, 37, 2416877). This comparison is therefore not justified in its current wording.

Answer

We thank the reviewer for this comment. Rather than directly comparing our results with the current state-of-the-art HgTe-based MWIR photodetectors, we have revised the manuscript to better contextualize the performance.

Specifically, we clarify that the detectivity achieved in this work is comparable to that reported for the first HgTe colloidal quantum dot MWIR photodetectors demonstrated in the early stage of the field (*Nature Photonics* 5, 489–493 (2011)). At the same time, we emphasize that our Ag₂Te photodetector significantly outperforms previously reported MWIR photodetectors based on non-toxic, heavy-metal-free quantum dots.

Based on this perspective, we have revised the manuscript text to highlight (i) the similar performance relative to early HgTe MWIR devices and (ii) the clear performance advantage over existing non-toxic MWIR quantum dot photodetectors. To further clarify the impact of our results, we compare our device with previous reports focusing on non-toxic materials and the spectral limits of Ag₂Te:

- InSb CQDs: Recently reported a D^* of 3.1×10^7 Jones at 3.5 μm (*Nano Lett.* 2025, 25, 13549).
- Ag₂Se CQDs (Intraband Transition): Showed a D^* of 7.8×10^6 Jones without cooling in the MIR region (*ACS Appl. Mater. Interfaces* 2021, 13, 49043).
- SnTe CQDs: Devices are shown to exhibit mA/W responsivity under MWIR flux; however, intrinsic limitations in D^* due to inverse photoresponse behavior (*ACS Sens.* 2018, 3, 2087).
- Ag₂Te CQDs Spectral Limit: While existing Ag₂Te has shown high D^* in shorter SWIR wavelengths (near the 1300 nm: 3×10^{12} Jones (*Nature Photonics* volume 18, pages 236–242 (2024)), 1700 nm: 9.0×10^{10} Jones (*ACS Materials Lett.* 2024, 6, 4988–4996)). To the best of our knowledge, the longest-wavelength detectivity previously reported for Ag₂Te was measured with a 2004 nm laser, yielding only 1.0×10^7 Jones (*Adv. Sci.* 2024, 11, 2407453).

In contrast, the Ag₂Te photodetector presented in this work successfully extends high-performance operation into the MWIR region, achieving a D^* of 1.2×10^9 Jones. While HgTe-based devices remain a benchmark for absolute performance, the transition toward non-toxic alternatives is increasingly important due to environmental and regulatory considerations, such as restrictions on hazardous substances (RoHS). In this context, our Ag₂Te photodetector provides a viable pathway toward high-performance, environmentally benign MWIR detection. We have incorporated this wavelength-dependent performance comparison into the revised manuscript and Supporting Information.

Initial version:

(line 239) The resulting specific detectivity (D^*) of 1.2×10^9 Jones at 78 K demonstrates competitive performance compared to HgTe photoconductor devices reported in previous reports^{45–48}.

Revised version:

(line 250) The device exhibited a specific detectivity (D^* , 1 kHz) of 1.2×10^9 Jones at 78 K under a bias of 0.02 V, with a measured responsivity of 1.9 mA/W for a device area of 2.6 mm². This D^* value is comparable to those reported for early HgTe CQDs MWIR photodetectors in 2011 (Nature Photonics 5, 489–493), indicating that the present device architecture maintains reliable MWIR detection performance despite employing a non-toxic material system. In addition, we summarize and compare the detectivity of previously reported non-toxic MWIR CQD photodetectors, as well as Ag₂Te CQD-based devices, in Supplementary Table 1. Within this context, the performance of our Ag₂Te photodetector demonstrates a clear advancement among non-toxic MWIR CQD systems reported to date.

Revised version (addition):

Table S1. Performance of the detectivity table of non-toxic MWIR and SWIR Ag₂Te photodetectors from previous studies.

Materials	Detectivity	Wavelength	Year	Ref
InSb	4.7×10^7 Jones	3.0 μm	2025	Nano Lett. 2025, 25, 13549
	3.1×10^7 Jones	3.5 μm		
SnTe	N/A	3.0 μm	2018	ACS Sens. 2018, 3, 10, 2087–2094
Ag₂Se	7.8×10^6 Jones	4.5 μm (Intraband)	2021	ACS Appl. Mater. Interfaces 2021, 13, 49043
Ag₂Te	3.0×10^{12} Jones	1.3 μm	2024	Nature Photonics volume 18, pages 236–242 (2024)
	9.0×10^{10} Jones	1.7 μm	2024	ACS Materials Lett. 2024, 6, 4988–4996
	1.0×10^7 Jones	2.0 μm	2024	Adv. Sci. 2024, 11, 2407453
	1.2×10^9 Jones	4.7 μm	2026	This Work

9. Several references are missing (e.g., at L45, L112, L224, L281). All statements that rely on prior work must be properly cited.

Answer:

We thank the reviewer for the comment about the missing reference. Following the advice, we added the reference as follows.

Initial version:

(line 45) This was followed by the development of phosphine-free synthetic methods,

(line 110) This is consistent with the results of Norris and his coworkers, showing that monoclinic Ag₂Te nanocrystals remain stable even when produced at 200 °C.

(line 224) Assuming $\nu_0 = 10^{12} \text{ s}^{-1}$, the Miller–Abrahams expression gives an average hopping rate $4.6 \times 10^9 \text{ s}^{-1}$,

(line 281) Such measured response times show competitive performance compared to previous reports. The measured fall time of 576 ns (Supplementary Table 1) is among the fastest reported for MWIR QD photodetectors, confirming the competitive temporal performance of this device.

Revised version:

(line 45) This was followed by the development of phosphine-free synthetic methods(*Nature Photonics* volume 18, pages236–242 (2024)),

(line 115) This is consistent with the results of Norris and his coworkers(J. Am. Chem. Soc. 2011, 133, 17, 6509–6512), showing that monoclinic Ag₂Te nanocrystals remain stable even when produced at 200 °C.

(line 236) The ν_0 is hopping attempt frequency (phonon frequency). Assuming $\nu_0 = 10^{12} \text{ s}^{-1}$ based on the reported paper (Phys. Rev. 1960, **120**, 745), the Miller–Abrahams expression gives an average hopping rate $4.6 \times 10^9 \text{ s}^{-1}$, corresponding to a mean hopping time of $\tau_{hop}(T) = 220 \text{ ps}$. (Taking $\nu_0 = 10^{13} \text{ s}^{-1}$ then the $\tau_{hop}(T) = 22 \text{ ps}$).

(line 305) Such measured response times show competitive performance compared to previous reports (Supplementary Table 1). The measured fall time of 523 ns is among the fastest reported for non-toxic MWIR QD photodetectors, confirming the competitive temporal performance of this device.

10. The Experimental Methods section is insufficiently detailed to allow reproducibility. For example, 1) exact precursor amounts and concentrations used for growing the large-size QDs are not mentioned at all! 2) ligands used for device fabrication are mentioned only as ‘shorter length’! 3) the device dimensions are not clearly written.

Answer:

We thank the reviewer for pointing out the need for additional experimental details to ensure reproducibility. We agree that the initially submitted Experimental Method section lacked sufficient specificity in several places. In response, we have substantially revised and expanded the Methods section to include all critical parameters required for reproducibility, as detailed below.

Initial version:

(line 233) Under cryogenic conditions, the noise level was measured to be 0.26 pA/Hz^{0.5} at 1000 Hz. Using this value, the specific detectivity was calculated with the following equation,

$$D^* = \frac{R\sqrt{A}}{i_n}$$

where R , A , and i_n are the responsivity, device area, and noise current density, respectively.

(line 238) The resulting specific detectivity (D^*) of 1.2×10^9 Jones at 78 K demonstrates competitive performance compared to HgTe photoconductor devices reported in previous reports⁴⁵⁻⁴⁸.

(line 439) **Synthesis of MWIR Ag₂Te**

The 10 mg/mL SWIR Ag₂Te solution (in oleylamine) was stirred under vacuum (<100 mTorr) for 1 h. After degassing, increase the reaction temperature to 180 °C, then rapidly inject 0.1 M silver nitrate (in oleylamine) and the reducing agent (Diphenylphosphine or Alane N,N-dimethylethylamine), and allow the reaction to proceed for 2–6 h.

Revised version:

(line 244) Under cryogenic conditions, the noise level was measured to be 0.26 pA/Hz^{0.5} at 1 kHz, as detailed in Supplementary Fig. 13. Using this value, the specific detectivity was calculated with the following equation,

$$D^* = \frac{R\sqrt{A}}{i_n}$$

where R , A , and i_n are the responsivity, device area, and noise current density, respectively.

(line 250) The device exhibited a specific detectivity (D^* , 1 kHz) of 1.2×10^9 Jones at 78 K under a

bias of 0.02 V, with a measured responsivity of 1.9 mA/W for a device area of 2.6 mm². This D* value is comparable to those reported for early HgTe CQDs MWIR photodetectors in 2011 (Nature Photonics 5, 489–493 (2011)), indicating that the present device architecture maintains reliable MWIR detection performance despite employing a non-toxic material system. In addition, we summarize and compare the detectivity of previously reported non-toxic MWIR CQD photodetectors, as well as Ag₂Te CQD-based devices, in Supplementary Table 1.

(line 455) **Synthesis of MWIR Ag₂Te**

Pre-formed SWIR Ag₂Te nanocrystals dispersed in OLAm (10 mg/mL) were used as the starting material. The nanocrystal solution was degassed under vacuum (<100 mTorr) for 1 h and subsequently heated to 180 °C under an inert atmosphere. Then, 0.1 mL of 0.1 M AgNO₃ solution in OLAm and 0.02 mL of reducing agent (DPP, Alane N,N-dimethylethylamine) were rapidly injected to initiate post-growth. An external reducing agent was added to weaken the bond between Ag⁺ ions and oleylamine on the surface of nanocrystals to lower the surface energy barrier and introduce additional silver. The reaction was maintained at 180 °C for 2-6 h, during which the nanocrystal size was tuned by varying the post-growth duration. Longer reaction times resulted in larger Ag₂Te nanocrystals and a corresponding red-shift of the absorption edge into the MWIR region (Supplementary Figs. 1-2). After completion, the reaction mixture was cooled to room temperature, and the products were purified by repeated precipitation and redispersion for further characterization and device fabrication.

Device Fabrication (Interdigitated Electrode, IDE)

A commercial interdigitated electrode (IDE) with Pt electrodes on insulating substrates was used for lateral photoconductor devices. As-synthesized MWIR Ag₂Te CQDs were dispersed in chlorobenzene at a concentration of ~100 mg/mL and deposited onto the IDE substrates by drop casting. After film deposition, a ligand exchange with shorter ligands was performed to improve charge transport within the CQD solid. Specifically, the CQD films were treated with a solution containing the CTAB ligand, followed by rinsing with IPA to remove excess ligand. The resulting CQD films uniformly covered the electrode area, forming an active channel between the interdigitated fingers. Electrical contacts were made directly through the pre-patterned Pt electrodes. The completed devices were loaded into an optical cryostat for temperature-dependent electrical and photocurrent measurements.

Device Fabrication (Vertical structure)

For vertical photodetector devices, sapphire with ITO electrodes (60 nm) on insulating substrates was first cleaned by sequential sonication in acetone and isopropanol, then dried under nitrogen. A film of MWIR Ag₂Te CQDs was deposited onto the ITO electrode by drop casting from a chlorobenzene solution (~100 mg/mL), followed by a post-ligands, identical to the procedure used for IDE devices. After CQD film formation, a MoO_x layer and Au top electrode were sequentially deposited by thermal evaporation to suppress dark current and enhanced infrared reflectivity. The final device structure was sapphire/ITO/MWIR Ag₂Te CQD/MoO_x/Au.

11. How is the size tunability with the reported synthetic method? The authors should include reaction conditions that control QD sizes, and corresponding size/absorption data.

Answer:

We thank the reviewer for the comment regarding size tunability and the corresponding optical response. We agree that a clear description of the growth parameters governing QD size, together with correlated size and absorption data, is essential.

In our synthetic approach, the Ag₂Te CQD size is primarily controlled by the post-growth reaction time at 180 °C in the presence of additional Ag precursor and a reducing agent. Increasing the post-growth duration systematically enlarges the nanocrystal size, resulting in a red-shift of the absorption edge from the SWIR to the MWIR regime.

The size-dependent absorption and structural evolution are explicitly provided in the Supporting Information. Specifically, Supplementary Figures S1 and S2 present FT-IR absorption spectra together with representative TEM images and corresponding size distributions for Ag₂Te CQDs with diameters ranging from approximately 4-9 nm (Fig. S1) to 10-11 nm (Fig. S2). These data demonstrate a clear correlation between increasing nanocrystal size and the progressive red-shift of the absorption peak from ~8280 cm⁻¹ to ~3000 cm⁻¹. Furthermore, Supplementary Figure S3 summarizes this size-bandgap relationship using a two-band k·p model, overlaid with the experimentally extracted size and absorption data. To improve clarity, we have revised the main manuscript to reference the Supporting Information figures when discussing size tunability explicitly and to clarify.

Initial version:

(line 93) Supplementary Figs. 1-2 show the TEM image of QDs synthesized, confirming a spherical morphology and a narrow size distribution (standard deviation < 12%). To confirm the size-dependent bandgap energy, we employed a two-band k·p approximation. (Supplementary Fig. 3).

Revised version:

(line 99) Supplementary Figs. 1 and 2 present representative TEM image of Ag₂Te QDs synthesized with varying post-growth reaction times, showing a spherical morphology and clear size evolution with prolonged growth. The associated absorption spectra reveal a consistent red-shift with increasing CQD size. The resulting size-dependent bandgap trend is further analyzed using a two-band k·p approximation, as shown in Supplementary Fig. 3.

12. The photocurrent spectra show dips around the region where C-H vibrational modes typically appear. Since the authors replaced long chain ligands with shorter ones, these features should not strongly affect the photocurrent. A clear explanation is necessary.

Answer:

We agree that ligand exchange with shorter ligands significantly reduces the contribution of long-chain organic ligands. However, it is worth noting that ligand exchange in colloidal quantum dot solids is rarely 100% complete, and a fraction of residual alkyl ligands may remain bound to the nanocrystal surface or trapped at interparticle interfaces.

Even a small population of residual C-H containing ligands can give rise to detectable vibrational signatures in mid-infrared photocurrent spectroscopy, particularly when combined with broadband thermal excitation and EVET processes. Therefore, the observed dips are attributed to a combination of extrinsic vibrational absorption, EVET, and possible residual ligand contributions.

Initial version:

(line 169) Their bandgap closely matches that of HgTe CQDs, which are the most representative MWIR CQDs in the Supplementary Fig. 9. Analysis of the photocurrent results in Figs. 2d and 2e, derived from temperature-dependent photocurrent spectra, show vibrational modes of CH₂ ligands and of atmospheric H₂O (3500–4000 cm⁻¹, O-H stretch) in ambient conditions.

Revised version:

(line 175) Their bandgap closely matches that of HgTe CQDs, a representative MWIR CQD (Supplementary Fig. 9). The temperature-dependent photocurrent spectra (Figs. 2d and 2e) exhibit features corresponding to vibrational modes of CH₂ ligands and atmospheric H₂O (3500-4000 cm⁻¹, O-H stretch) measured under ambient conditions. While ligand exchange with shorter ligands was employed, complete removal of all alkyl species cannot be strictly assumed, and residual vibrational contributions may therefore modulate the MWIR photocurrent response.

13. The rise and fall time calculations appear optimistic. The photocurrent does not reach saturation at the reported modulation frequencies, which makes the extracted values unreliable. A dB vs frequency (Hz) plot is necessary to determine the device bandwidth and speed.

Answer:

We appreciate the reviewer's insightful comment on the non-ideal modulation waveform and its implications for the device's response time.

First, the photoresponse of the IDE-based photodetector was investigated using a QCL combined with an optical chopper. Because the QCL operates in a pulsed mode, the pulse repetition frequency was fixed at its maximum value of 625 kHz, while the chopper modulation frequency was varied from 20 Hz to 1 kHz. Under pulsed excitation, the measured signal represents the device response to a train of optical impulses rather than to an ideal square-wave modulation. To enable a clearer physical interpretation of the response dynamics, both the step-modulated transient response (Used chopper) and the frequency-dependent normalized photoresponse.

1. Step-modulated transient response (Used chopper)

Under chopper-modulated illumination at 1000 Hz, the device exhibits a well-defined square-wave photoresponse with a fall time of 124 μs . As shown in the figures below and in the Supporting Information, this square response is consistently preserved over the frequency range from 100 Hz to 1000 Hz.

(MWIR Ag₂Te on IDE, 78 K, 2900 cm⁻¹, 0.1 V)

Under illumination from the QCL at 2900 cm⁻¹, the device exhibited clear on-off photoresponses at a modulation frequency of 1 kHz, with a rise time of 141 μs and a fall time of 124 μs .

2. Frequency-dependent Photoresponse (Direct QCL Modulation)

To further evaluate the dynamic bandwidth, frequency-domain measurements were performed by directly tuning the QCL pulse-repetition frequency from 10 Hz to 500 kHz. The photoresponse as a function of modulation frequency is shown below.

From the normalized frequency response, a -3 dB cutoff frequency of 3241 Hz was extracted. This value is in good agreement with the cutoff frequency estimated from the measured fall time using the following relation:

$$\text{Fall-time and bandwidth relationship: } f_{3\text{ dB}} \approx \frac{0.35}{\tau_{\text{fall}}} = 2822\text{ Hz } (\tau_{\text{fall}}: 124\ \mu\text{s})$$

We confirmed consistency between the time-domain transient response and the frequency-domain bandwidth, and the data were prepared for the manuscript and SI.

Furthermore, to more clearly distinguish the photoresponse duration and enhance the reliability of response-speed evaluation in the vertical device configuration, we measured the pulse-frequency-dependent photoresponse intensity. This approach enables a robust, quantitative characterization of the photodetector's temporal behavior and operational bandwidth.

Accordingly, we have extensively revised the contents of Figures 3 and 4 to provide a more comprehensive analysis.

Initial version:

(line 228) These findings confirm the ability for rapid modulation, as shown in Fig. 3d. Clear on-off responses were observed at modulation frequencies from 100 Hz to 1 kHz under a 600 °C blackbody radiation. This can be attributed to the efficient generation and rapid extraction of carriers under illumination.

(line 279) Under photoexcitation by a quantum cascade laser at 2800 cm^{-1} ($3.6 \mu\text{m}$), modulated at 625 kHz, the as-fabricated vertical-type Ag_2Te photodetector exhibited a nanosecond-scale photoresponse, with a rise time of 230 ns and a fall time of 576 ns. Such measured response times show competitive performance compared to previous reports.

Revised version:

Fig. 3| Device performance of MWIR Ag_2Te photodetectors. (Interdigitated Electrode, IDE) a, J–V characteristics measured at various temperatures. (inset: responsivity versus operating temperature).

b, Activation energy extracted from temperature-dependent measurements. **c**, Conductance G as a function of $T^{-1/4}$. **d**, Temporal photoresponse at 1 kHz under 2900 cm^{-1} laser by Quantum Cascade Laser (QCL). **e**, The noise current density recorded under dark conditions. **f**, Time-dependent current curve at 0.1 V derived from blackbody radiation at 20, 37, 40 $^{\circ}\text{C}$. **g**, Schematic structure of the MWIR Ag_2Te device integrated on the IDE. **h**, Infrared thermal imaging setup and corresponding images confirm the photoresponse of the MWIR Ag_2Te device.

(line 240) These findings confirm the ability for rapid modulation, as shown in Fig. 3d and Supplementary Fig. 12. Under illumination from the QCL at 2900 cm^{-1} , the device exhibited clear on-off photoresponses at a modulation frequency of 1 kHz, with a rise time of 141 μs and a fall time of 124 μs . This can be attributed to the efficient generation and the rapid extraction of carriers under illumination.

Fig. 4| Device performance of MWIR Ag_2Te photodetectors. (Vertical structure) a, Schematic of MWIR Ag_2Te photoconductor device and a cross-sectional focused ion beam (FIB) image **b**, On-off characteristics with a 2900 cm^{-1} mid-infrared QCL. The pulse duration was 625 kHz. **c**, The response time of the MWIR Ag_2Te QD photodetector plotted with those of a commercial MCT bulk crystal photodetector (Blue, Kolmar Tech., KMPV11-1-J1) and a commercial InSb bulk crystal photodetector (Grey, Infrared Associates, 1176-2C-14-0.5). **d**, Time-domain photocurrent responses measured under QCL pulse excitation with repetition frequencies ranging from 5 Hz to 500 kHz. **e**, Corresponding normalized photoresponse as a function of modulation frequency, exhibiting a -3 dB cutoff frequency of 69.6 kHz.

(line 301) The vertical device exhibits a measurable photoresponse under blackbody illumination at 873 K, indicating effective infrared photodetection (Supplementary Fig. 14). Under photoexcitation by

a QCL at 2900 cm^{-1} ($3.4\ \mu\text{m}$), modulated at 200 kHz, the as-fabricated vertical-type Ag_2Te photodetector exhibited a nanosecond-scale photoresponse, with a rise time of 211 ns and a fall time of 523 ns. Such measured response times show competitive performance compared to previous reports (Supplementary Table 1).

(line 317) To assess the robustness of the device response under repetitive optical excitation, pulse-frequency-dependent photoresponse measurements were performed over a modulation range from 5 Hz to 500 kHz (Fig. 4d). The photocurrent amplitude gradually decreased with increasing repetition frequency, reflecting the device's finite response dynamics. Fig. 3e shows the normalized frequency response yields a -3 dB cutoff frequency of 69.6 kHz. This frequency-domain analysis provides independent validation of the high-speed temporal response, consistent with the trends observed in transient pulse measurements.

Revised version (addition):

Figure S12. Temporal and frequency-dependent photoresponse of the device. (a) Time-domain photoresponse measured under optical chopper modulation, showing distinct on–off switching behavior. (b) Pulse-frequency-dependent photoresponse measured by directly modulating the QCL pulse repetition frequency, illustrating the evolution of the response amplitude as a function of pulse frequency.

From the normalized frequency response, a -3 dB cutoff frequency of 3.2 kHz was extracted. This value is in good agreement with the cutoff frequency estimated from the measured fall time using the following relation:

$$\text{Fall-time and bandwidth relationship: } f_{3\text{ dB}} \approx \frac{0.35}{\tau_{fall}} = 2.8\text{ kHz } (\tau_{fall}: 124\ \mu\text{s})$$

The bandwidth derived from the pulse measurements is in good agreement with the data shown in Fig. 3d.

Figure S14. (a) Current density–voltage (J-V) characteristics of the device measured in the dark and under illumination. The net photocurrent is shown, calculated as the difference between the two curves. (b) On–off photoresponse was measured using an optical chopper, while the intrinsic response time was evaluated using a pulsed laser because the chopper’s modulation frequency was limited to 1 kHz. (c) Responsivity was calculated from current and power densities. The device’s active area is 1.05 mm². At 1 V (positive bias), the responsivity is approximately 40 mA/W. The lower responsivity relative to the IDE structure is attributed to the ITO electrode’s low infrared transmittance.

14. The authors report noise only at 1 kHz. However, low-frequency noise (1/f noise) is essential for evaluating detector performance. This must be included

Answer:

We appreciate the comments. While noise spectra are reported at 1 kHz, low-frequency noise components such as 1/f noise and generation–recombination noise are known to dominate in CQD-based infrared photodetectors and critically impact practical detectivity, particularly for low-bandwidth imaging applications. Following your comment, we have added the data to the supporting information.

Initial version:

(line 233) Under cryogenic conditions, the noise level was measured to be $0.26 \text{ pA/Hz}^{0.5}$ at 1000 Hz. Using this value, the specific detectivity was calculated with the following equation,

Revised version:

(line 244) Under cryogenic conditions, the noise level was measured to be $0.26 \text{ pA}/\sqrt{\text{Hz}}$ at 1 kHz, as detailed in Supplementary Fig. 13. Using this value, the specific detectivity was calculated with the following equation,

$$D^* = \frac{R\sqrt{A}}{i_n}$$

where R , A , and i_n are the responsivity, device area, and noise current density, respectively.

Revised version (addition):

Figure S13. The noise spectral density of the Ag₂Te photodetector was measured under dark conditions. At low frequencies (below ~100 Hz), the noise exhibits a characteristic 1/f dependence. As the frequency increases, the noise spectrum transitions to a frequency-independent region above ~100 Hz, corresponding to a white-noise-dominated regime.

Based on the dark current and the dynamic resistance measured at a bias of 0.02 V, the theoretical noise contributions were estimated.

The shot-noise-limited current noise: $0.23 \text{ pA}\sqrt{\text{Hz}}$, $i_{shot} = \sqrt{2qI\Delta f}$

The Johnson–Nyquist thermal noise: $0.19 \text{ pA}\sqrt{\text{Hz}}$, $i_{theraml} = \sqrt{4k_B R \Delta f}$

The experimentally measured noise density of $0.26 \text{ pA}\sqrt{\text{Hz}}$ at 1 kHz shows close agreement with these theoretical limits. This correspondence indicates that at a modulation frequency of 1 kHz, low-frequency 1/f noise is effectively suppressed and the device operates near its intrinsic noise floor.

For comparison, the instrument noise floor of the measurement system was independently measured by shorting the input of the current preamplifier to ground. The resulting background noise level is significantly lower than the noise measured with the device across the entire frequency range of interest, including at 1 kHz. This confirms that the noise used for detectivity evaluation is dominated by the intrinsic device noise rather than by the measurement electronics.

15. The reported NETD of 300 mK is somewhat high for an MWIR photodetector. The authors used a blackbody source to image the letter “K,” but a blackbody emits across the entire spectrum from visible to far-IR. The manuscript must specify the blackbody temperature and indicate whether any filters were used to block the SWIR/NIR/visible components.

Answer:

We thank Reviewer 3 for the comment. We agree with the Reviewer’s observation regarding the NETD value. While an NETD of 300 mK is somewhat higher than that of state-of-the-art MWIR photodetectors based on toxic (e.g., HgTe) or epitaxial (e.g., InSb) materials, it is important to evaluate this performance in the context of non-toxic, solution-processed material platforms. In this category, our device achieves an order-of-magnitude higher responsivity than previously reported non-toxic MWIR photodetectors (Table S1). Since NETD is inversely proportional to the signal-to-noise ratio and responsivity, this enhancement represents a significant milestone for environmentally friendly infrared imaging.

During the imaging experiment of the “K” pattern, a Ge window, the long-pass filter, was placed in front of the cryostat window. The Ge window blocks radiation above 5000 cm^{-1} , thereby ensuring that the device responds only to the SWIR and MWIR spectral components emitted from the blackbody source.

The blackbody temperature was set to 1200 K using a Thorlabs SLS303 source. We have added these experimental details to the revised manuscript for clarity.

Initial version:

(line 257) The blackbody radiation passing through the mask’s hole was projected onto the fabricated detector, and the photocurrent generated by the pixel was recorded to create the infrared image.

Revised version:

(line 274) A Ge window, the long pass filter, was placed in front of the cryostat to block radiation with wavenumber above 5000 cm^{-1} , ensuring that the detector responded only to the SWIR and MWIR spectral components of the blackbody emission. The blackbody temperature was set to 1200 K (Thorlabs SLS303)

16. If quantum confinement exists across the MWIR range, the PL emission peak should shift systematically with particle size. Is it possible to show this?

Answer:

We thank Reviewer 3 for the comment. Because photoluminescence is significantly quenched by C–H vibration–induced EVET near 2900 cm^{-1} , direct PL-based analysis of size-dependent spectral shifts is not reliable. Instead, we investigated the particle-size-dependent photocurrent spectroscopy, which clearly shows the red-shift from 2.3 to $4.7\ \mu\text{m}$. (Figure 2).

17. Finally, the IDE device is fabricated on a glass substrate. However, glass absorbs strongly in the mid-IR and can generate thermal contributions to the photocurrent. It would be more appropriate to use a substrate such as CaF_2 or sapphire for MWIR measurements.

Answer:

We thank Reviewer 3 for the comment on the direction of light incidence into the device. For the vertically structured devices, the photodetectors were fabricated on sapphire substrates; therefore, the incident light passed through the sapphire/ITO electrode stack before reaching the CQD active layer. In contrast, for the IDE-based devices, the light was directly irradiated onto the CQD layer without passing through any glass or substrate. Thus, the optical transmittance of the substrate need not be considered. We mentioned this in the Method section at question (10) and in Fig. 3g.

Initial version:

Revised version:

REVIEWER COMMENTS

Reviewer #1 (Remarks to the Author):

The authors have revised the manuscript, and the reviewer would like to recommend acceptance of the current version.

Answer:

We sincerely thank Reviewer 1 for his/her time, effort, and comments on our manuscript. It helped us improve our manuscript.

Reviewer #2 (Remarks to the Author):

One final minor comment related to the original question #4. "In Figure 3a, the legend currently labels all curves as being measured "under light." However, the dotted curves appear to correspond to the dark measurements. The authors should confirm this and correct the legend accordingly."

In response to this comment, the authors added new Figure S10(c) to present the true dark condition I-V. The optical power of 873K blackbody in the 3-5 MWIR band is about 10^4 times larger than that of the 300K black body. **Hence it is somewhat surprising to see that current density of '78K, Light 298K' data jumps to a similar level to '78K, Light 873K data'** (i.e. a much larger separation between these two curves would typically be expected).

The Figure S10(c) looks more like a LWIR detector rather than the MWIR detector. There might be more complex physics related to this device behavior. Nevertheless, while this is intriguing and may warrant further clarification in future studies, **it does not fundamentally alter the main claims or conclusions drawn in the manuscript.**

Overall, the authors have satisfactorily addressed the previously raised comments through clarified figures, additional supporting data, expanded discussion, and appropriate references. **The manuscript appears ready for publication in its present form.**

Answer:

We thank Reviewer 2 for the insightful and positive comment. We understand that, when comparing the dark current values at 78 K, 298 K, and 873 K in the Figure with the logarithmic scale, the temperature dependence might look stronger at lower temperatures for Figure S10(c) in our manuscript. However, when the same data are plotted on a linear scale, the differences are not that great, as shown below.

The temperature dependence of the dark current would not be direct evidence of an LWIR detection. The experimentally measured photocurrent spectrum clearly shows a response in the MWIR region (Figure 2), confirming that the bandgap transition of Ag₂Te QD in the MWIR is responsible for detection.

Reviewer #3 (Remarks to the Author):

The authors have taken my comments into consideration, they have amended several of the previous claims that they could not be supported by the data and now the MS is technically more rigorous. I feel more comfortable now with its publication.

There is however a subtle point that requires further attention:

(1) the authors mention responsivity of 1.1 A/W in the abstract and at several points in the manuscript, but when they refer to D^* conditions the responsivity is only 1.9 mA/W, i.e. 3 orders of magnitude lower...is this because of the lower applied bias?

For sure the abstract needs to be amended because as written it implies that R of 1.1A/W and D^* of 1.2×10^9 Jones are simultaneously achieved but it seems that is not the case, so it is somewhat misleading.

Answer:

We sincerely thank Reviewer 3 for the comments. To clarify, we revised the responsivity value in the Abstract to fully align with the data presented in the main manuscript. To show both responsivity and corresponding D^* , we show the range of R values measured at various applied bias and the D^* of 1.2×10^9 Jones measured at 0.02 V. The revised abstract now includes all information as shown below.

Revised version:

Abstract

As demand for sustainable and biocompatible technologies grows, low-toxicity mid-infrared materials, such as silver chalcogenides, have attracted significant interest. Herein, we report mid-wavelength infrared (MWIR) tunable Ag_2Te colloidal quantum dot (CQD) through a post-growth method starting from short-wavelength infrared Ag_2Te CQDs. Using the synthesized MWIR Ag_2Te CQDs, we successfully fabricated a MWIR photodetector covering the full MWIR spectral range (3–5 μm) with an onset wavelength extending to 6.9 μm . At 78 K, the photodetectors exhibit a photoresponse time of 230 ns (rise) and 576 ns (fall). Responsivity varies from 1.9×10^{-3} A/W at 0.02 V to 1.1 A/W at 0.5 V, depending on the applied bias, and the specific detectivity (D^*) we measured from the device is 1.2×10^9 Jones at 0.02 V. The measured noise-equivalent temperature difference (NETD) of 0.3 K enables us to reliably distinguish temperature variations between 37 °C and 40 °C, directly enabling the diagnosis of fever-level body temperatures.

(2) Can the authors show a bias dependent R, D^* ? Also is the spectrum of EQE applied bias invariant?

Answer:

We further added bias-dependent responsivity to Figure S10. Yes. The photocurrent spectrum in the manuscript was measured at an invariant bias.

Revised version:

Figure S10